# Vertical redistribution of moisture and aerosol in orographic mixed-phase clouds

Annette K. Miltenberger[1,2], Paul R. Field[1,3], Adrian H. Hill[3], and Andrew J. Heymsfield[4]

[1]Institute of Climate and Atmospheric Science, School of Earth and Environment, University of Leeds, United Kingdom
[2]Institute for Atmospheric Physics, Johannes Gutenberg-University Mainz, Germany
[3]MetOffice, Exeter, United Kingdom
[4]National Center for Atmospheric Research, Boulder, Colorado

**Correspondence:** Annette K. Miltenberger (amiltenb@uni-mainz.de)

**Abstract.** Orographic wave clouds offer a natural laboratory to investigate cloud microphysical processes and their representation in atmospheric models. Wave clouds impact the larger-scale flow by the vertical redistribution of moisture and aerosol. Here we use detailed cloud microphysical observations from the ICE-L campaign to evaluate the recently developed Cloud Aerosol Interacting Microphysics (CASIM) module in the Met Office Unified Model (UM) with a particular focus on different parameterisations for heterogeneous freezing. Modelled and observed thermodynamic and microphysical properties agree very well (deviation of air temperature $< 1\,\mathrm{K}$, specific humidity $< 0.2\,\mathrm{g\,kg^{-1}}$, vertical velocity $< 1\,\mathrm{m\,s^{-1}}$, cloud droplet number concentration $< 40\,\mathrm{cm^{-3}}$), with the exception of an overestimated total condensate content and a too long sedimentation tail. The accurate reproduction of the environmental thermodynamic and dynamical wave structure enables the model to reproduce the right cloud in the right place and at the right time. All heterogeneous freezing parameterisations except Atkinson et al. (2013) perform reasonably well, with the best agreement in terms of the temperature dependency of ice crystal number concentrations for the parameterisations of DeMott et al. (2010) and Tobo et al. (2013). The novel capabilities of CASIM allowed testing of the impact of assuming different soluble fractions of dust particles on immersion freezing, but this is found to only have a minor impact on hydrometeor mass and number concentrations.

The simulations were further used to quantify the modification of moisture and aerosol profiles by the wave cloud. The changes in both variables are on order of $15\,\%$ of their upstream values, but the modifications have very different vertical structures for the two variables. Using a large number of idealised simulations we investigate how the induced changes depend on the wave period ($100 - 1800\,\mathrm{s}$), cloud-top temperature ($-15$ to $-50\,^\circ\mathrm{C}$) and cloud thickness ($1 - 5\,\mathrm{km}$) and propose a conceptual model to describe these dependencies.

# 1 Introduction

The advent of (sub-)kilometre scale numerical weather prediction models in recent years has strongly improved the prediction of clouds and precipitation (e.g. Clark et al., 2016). However, simplification in the representation of cloud microphysical processes and incomplete physical understanding of some key processes result in fairly large uncertainties in the representation of individual cloud microphysical processes, which also impact the macroscopic appearance of clouds, precipitation formation and cloud evolution (e.g. Muhlbauer et al., 2010; Johnson et al., 2015). To improve the representation of cloud microphysical processes and to reduce the associated uncertainty, the combination of model simulations with detailed observational data from dedicated field campaigns is of fundamental importance alongside the careful investigation of individual processes in the laboratory. Clouds forming in laminar flow in the vicinity of significant topography, so-called orographic wave clouds, have been suggested as natural laboratories to investigate cloud processes under ambient atmospheric conditions (e.g. Heymsfield and Miloshevich, 1993; Field et al., 2001; Muhlbauer and Lohmann, 2009). In contrast to convective cloud fields, the quasi-stationary, laminar flow provides well-constraint thermodynamic environment and dynamic forcing and allows for direct comparisons between observations and model results (e.g. Heymsfield and Miloshevich, 1993; Eidhammer et al., 2010; Field et al., 2012).

Orographic clouds known to be important for weather and climate, as they occur frequently in mountainous regions (e.g. Grubisic and Billings, 2008; Vosper et al., 2013), modify regional precipitation patterns (e.g. Sawyer, 1956; Smith et al., 2015) and influence radiative fluxes (e.g. Joos et al., 2008). Most studies on orographic clouds have focussed on their contribution to surface precipitation and its distribution, which has been investigated in a large number of idealised and realistic simulations with models of various complexity (e.g. Houze, 2012; Miltenberger et al., 2015; Henneberg et al., 2017). It has been shown that depending on the upstream conditions and the shape of the topography different cloud microphysical processes dominate the precipitation formation (e.g. Jiang and Smith, 2003; Colle and Zeng, 2004) and varying ambient aerosol concentrations can modify precipitation amounts and patterns (e.g. Muhlbauer et al., 2010; Zubler et al., 2011; Xiao et al., 2015). Precipitation formation does not only result in a vertical redistribution of moisture, but also a vertical transport of aerosol particles, which are incorporated into hydrometeors during cloud droplet or ice crystal nucleation (nucleation scavenging) or by aerosol-hydrometeor collisions (impaction scavenging) (e.g. Xue et al., 2012; Pousse-Nottelmann et al., 2015). However, not all aerosol particles incorporated into hydrometeors are removed to the surface, as a significant fraction of condensate evaporates before reaching the ground and the associated aerosol particles are released upon evaporation (or sublimation) (e.g. Xue et al., 2012; Pousse-Nottelmann et al., 2015). This results in modifications of the vertical profile of aerosol number and also the aerosol chemical composition. These changes modify the precipitation formation in clouds that form later in the same airmass, although to a lesser extend than varying upstream humidity of aerosol number concentration (Xue et al., 2012).

While orographic clouds producing (large) amounts of precipitation are very relevant in socio-economic terms, isolated wave clouds in the middle troposphere, which do not produce surface precipitation, are better suited to study the basic mixed-phase cloud processes of heterogeneous freezing, depositional growth, hydrometeor sedimentation and aerosol transport. In contrast to thicker orographic clouds, the collision-coalescence process is less important and the interactions between air parcels

travelling through the clouds at different altitudes is minimal. Also, their smaller horizontal and vertical extent implies that representative observations are obtained more easily. One particular question, for which observations in isolated mid-tropospheric mixed-phase wave clouds has been instrumental, is the glaciation of clouds. The formation of ice in all mixed-phase clouds, not only orographic wave clouds, plays a crucial role for the efficiency of precipitation formation (as already pointed out in early studies by Bergeron (1935) and Findeisen (2015)) and the cloud optical properties (e.g. Joos et al., 2014; Vergara-Temprado et al., 2018).

In the atmosphere ice forms either via homogeneous freezing of solution droplets at temperature colder than about $-35\,^\circ$C or at warmer temperatures through the mediation of certain aerosol particles, which are called ice nucleating particles (INP). INP can trigger ice formation via different processes, including immersion, contact and deposition freezing (e.g. Kanji et al., 2017). Aircraft observations in orographic wave clouds have demonstrated the large increase in ice crystal number concentration due to the onset of homogeneous freezing at cold cloud top temperatures: For example, Heymsfield and Miloshevich (1993) showed that ice crystal concentrations of $\sim 60\,\mathrm{cm}^{-3}$ observed at temperatures colder than $-35\,^\circ$C in wave clouds over the mountain states of the United States are consistent with parcel-model predictions assuming homogeneous freezing. Ice crystal concentrations at warmer temperatures were below the detection limit of the particle probes. Similarly, for wave clouds over Scandinavia Field et al. (2001) found homogeneous freezing to be dominant at temperatures colder than $-35\,^\circ$C, while ice at warmer temperatures was most likely formed via immersion or contact nucleation, i.e. freezing mechanisms requiring INPs. Ice crystal number concentrations at these warmer temperatures has been observed to correlated with the presence of large aerosol particles (Baker and Lawson, 2006; Eidhammer et al., 2010) and chemical analysis of ice crystal residuals found predominantly mineral dust with some contributions from organics and salts, which are known to be efficient INPs (e.g. Targino et al., 2006; Pratt et al., 2010). Depending on whether INPs are incorporated before or during the freezing event, different heterogeneous freezing mechanisms are distinguished. In mixed-phase orographic clouds immersion freezing, i.e. INPs acting first as cloud condensation nuclei and later initiating the freezing of the cloud droplets, is likely the dominant freezing mechanism according to model-based (Hande and Hoose, 2017) analysis and comparison between parcel model simulations and observations (Field et al., 2001; Eidhammer et al., 2010). Deposition and contact freezing are likely not important. However, Cotton and Field (2002) could not completely reconcile box-model simulations using known freezing mechanisms with observations of hydrometeor number concentrations and mass mixing ratios.

The representation of heterogeneous freezing in numerical models relies on empirical relationships involving aerosol number concentrations and temperatures, because the fundamental processes of the ice nucleation process and those determining the efficiency of specific aerosol particles to act as INP are not yet understood. Several empirical formulation of heterogeneous (immersion) freezing have been proposed: Early parameterisations such as Fletcher (1958) or Meyers et al. (1992) are solely based on ambient air temperature, while later parameterisations additionally take into account the number concentration of large ($> 0.5\,\mu$m) aerosol particles (e.g. DeMott et al., 2010; Tobo et al., 2013; DeMott et al., 2015). The main difference between the latter parameterisations is the geographic regions, in which the underlying observations were made, and hence they likely represent different chemical and/or mineralogical compositions of the INP population. Other recent parameterisations use estimates of the temperature-dependent number of active sites on specific materials and the surface area of the aerosol

population to predict the number of INPs at a given temperature (Niemand et al., 2012; Atkinson et al., 2013). Again the main difference between the parameterisations is the materials, for which the number of actives sites was determined. It is not clear how the different parameterisations affect cloud properties and whether the difference between the parameterisations can be directly assessed with observations of ice crystal number concentrations.

Previous work has demonstrated the usefulness of observations in orographic clouds to investigate cloud microphysical processes. However, detailed cloud-microphysical analysis in models was limited to parcel, column or idealised two-dimensional simulations. Here we use observations in isolated, mid-level wave clouds during the ICE-L campaign to assess the performance of the recently developed Cloud-Aerosol Interacting Module (CASIM) in three-dimensional simulations with Met Office Unified Model (UM), i.e. a non-hydrostatic model used for operational weather prediction. The objectives of the present work are
in particular:

- – Is the numerical weather prediction model able to capture the thermodynamic conditions and wave cloud dynamics with sufficient accuracy, i.e. i.e. the right cloud in the right place at the right time, to allow for a direct comparison of cloud microphysical properties between model and observations?

- – Can observations of the vertical variation in ice crystal number concentration be used to assess the validity of different
heterogeneous freezing parameterisations?

- – How large is the modification of the water vapour and aerosol profiles by the wave cloud? How does the downward transport of water vapour and aerosols depend on the upstream thermodynamic conditions? And under which conditions is the downward flux largest, i.e. can be best observed in future campaigns?

The analysis focusses on a wave cloud over the Central United States probed with the National Science Foundation (NSF)
C-130 aircraft (16[th] November 2007, RF03) during the Ice in Clouds Experiment — Layer Clouds (ICE-L) (Heymsfield et al., 2011; Pratt et al., 2010). Data from ICE-L have been used to investigate the relationship between upstream INP measurements and ice crystal number concentrations (Eidhammer et al., 2010; Field et al., 2012), the depositional growth of ice crystals (Heymsfield et al., 2011) and to investigate the impact of using adaptive ice crystal habits in idealised model simulations (Dearden et al., 2012). The chemical analysis of cloud droplet and ice crystal residuals by Pratt et al. (2010) indicated that
INPs active in the observed wave clouds are most likely mineral dust internally mixed with a significant salt component, as may be expected from aerosols emitted from playas in the Central United States.

Details on the observations, models and their set-up are provided in the following section. In section 3 we present the comparison of observed wave cloud properties to the results from high-resolution simulations with the Met Office UM with a specific focus on the vertical gradient in ice crystal number concentration (sec. 3.3. A Lagrangian analysis of the simulations provides
insight into the modification of humidity and aerosol profiles by the wave cloud (section 4.1). The dependence of amplitude and shape of this modification on the gravity wave length and upstream thermodynamic conditions determining cloud top temperature and cloud thickness is assessed with additional idealised simulations in section 4.2. Finally, section 5 summarises the results and discusses implications for future aircraft observations in orographic wave clouds to constrain mixed-phase cloud

microphysics.

## 2  Data and Methods

### 2.1  Observational data from ICE-L

We use data from various instruments onboard of the National Science Foundation (NSF) C-130 aircraft for information of aerosol, cloud and ice populations in the mixed-phase clouds observed in RF03 of the ICE-L campaign. Details on the
instrumentation can be found in Heymsfield et al. (2011) for hydrometeor and aerosol size distributions and Pratt et al. (2010) for the aerosol chemical composition. Here we focus on a wave cloud observed on $16^{th}$ November 2007 (RF03), for which observations from three different altitudes are available within a time interval of roughly $40 \, min$. For the model evaluation, we use the King liquid water probe for total liquid water content, the 2D-C and the 2D-S probe for ice number concentrations and estimated ice water content, the CDP (cloud droplet probe) for cloud droplet number concentrations, the tuneable diode laser
hygrometer (TDL) for humidity measurements, the counterflow virtual impactor (CVI) for total water content, and aerosol size distributions from the Ultra High Sensitivity Aerosol Spectrometer for size-resolved number concentrations (UHSAS). The data from 2D-C and 2D-S are restricted to particles larger than $50 \, \mu m$. The small threshold particle size for the 2D-C is justified by the good agreement between ice crystal number concentrations from the 2D-S and 2D-C. Shattering in wave clouds is very likely not a large issue due to the predominantly small size of the ice crystals. As in Field et al. (2012) we correct the
TDL humidity such that it is consistent with water saturation in the regions with a liquid water content (from the King liquid water probe) larger than $0.02 \, g \, m^{-3}$. Further details on the data and its post-processing can be found in Field et al. (2012).

### 2.2  The Unified Model

We use the Unified Model (UM), the numerical weather prediction model developed by the MetOffice and used for operational forecasting in the UK, to conduct simulations of the wave cloud observed during research flight 3 of the ICE-L campaign
($16^{th}$ November 2007). A global simulation (UM vn10.8, GA6 configuration, N512 resolution, Walters et al. (2017)) starting from the operational analysis at 12 UTC on $16^{th}$ November 2007 provides the initial and lateral boundary conditions for regional model simulations. Two regional nests are used, the first with a horizontal grid spacing of $1 \, km$ and the second with a grid spacing of $250 \, m$. Both nests are centred at the location of the observed wave cloud ($42.12 \, °N$, $-105.10 \, °E$). The analysis presented in this paper focuses on the innermost nest. In the vertical we use a stretched vertical coordinate system
with 140 levels, which provides a vertical resolution of $130 - 200 \, m$ at the altitude of the observed cloud. Mass conservation is enforced in the regional simulations (Aranami et al., 2014, 2015) and sub-grid scale turbulent processes are represented with a 3D Smagorinsky-type turbulence scheme (Halliwell, 2015; Stratton et al., 2015). The cloud microphysics are represented with the Cloud-AeroSol Interacting Microphysics (CASIM) module (see section 2.4). As we are particularly interested in the impact of ice nucleating particles (INPs) in the cloud we conduct sensitivity experiments with different heterogeneous ice

nucleation parameterisations as well as different assumptions regarding the incorporation of ice nucleating particles (INP) into cloud droplets, which is pre-requisite for immersion freezing. The details of these sensitivity experiments are described in section 2.4.

## 2.3 The KiD Model

For the analysis of a large set of wave clouds we conduct additional idealised simulations with the Kinematic Driver Model (KiD, Shipway and Hill (2012); Hill et al. (2015)). The KiD model uses prescribed dynamics to drive different microphysics modules and hence testing of different cloud microphysics and flow configurations in a relatively simple framework. Here, we conduct two-dimensional simulations of wave clouds with different horizontal wavelength (period T between $100\,\mathrm{s}$ and $1800\,\mathrm{s}$), cloud top temperature ($\mathrm{t_{ct}}$ between $-12\,^\circ\mathrm{C}$ and $-50\,^\circ\mathrm{C}$) as well as with different cloud thickness ($\mathrm{z_c} = \mathrm{z_{ct}} - \mathrm{z_{cb}}$ between $1000\,\mathrm{m}$ and $4000\,\mathrm{m}$). This results in a total of 2268 simulations with different flow and/or thermodynamic conditions. All simulations are carried out with a vertical resolution of $50\,\mathrm{m}$, 200 vertical levels and a time-step of $1\,\mathrm{s}$.

At each model level a vertical velocity time series is prescribed:

$$w(t,z) = A \cdot T^{-1} \sin\left(2\pi t \cdot T^{-1}\right) \tag{1}$$

with $A = 2880\,\mathrm{m}$. Multiple simulations are carried out with $T \in [100, 1800]\,\mathrm{s}$. This formulation leads to a maximum vertical displacement of $\eta = A\pi^{-1} \approx 916.7\,\mathrm{m}$ irrespective of the chosen period T. This value of $\eta$ corresponds to the mean maximum vertical displacement of trajectories derived from the UM simulation, which pass through the wave cloud. The vertical velocity is set to zero after T. The time period T controls the horizontal extend of the wave cloud. Using typical horizontal wind speeds of between $10 - 30\,\mathrm{m\,s^{-1}}$ the sampled T range translates into along flow cloud extend between $1 - 54\,\mathrm{km}$. This covers the range of wavelength found in climatological studies of wave cloud (e.g. Grubisic and Billings, 2008). Although these climatological studies focus on lee wave clouds and to our knowledge no climatology of cap clouds is available, this range should be representative of the isolated mid-level wave clouds that are the focus of the present study. Note, that orographic clouds responsible for orographic precipitation typically have a much larger horizontal extend, at least if they do not form at isolated hills or mountains. Further note, that the wavelength cited above only pertain to the thermodynamic constraints for cloud formation. In the case of hydrometeors a finite evaporation timescale, the cloud can have a longer spatial extend (also in our KiD simulations).

The upstream temperature profiles is given by a lapse rate of $-8.104 \cdot 10^{-3}\,\mathrm{K\,m^{-1}}$ and a surface temperature of $32.1\,^\circ\mathrm{C}$. The initial pressure profile is computed using the hydrostatic approximation with a pressure of $886.2\,\mathrm{hPa}$ at $1000\,\mathrm{m}$ altitude (lowermost level). An initial profile of relative humidity is used with a relative humidity of $45\,\%$ below the moist layer, $70\,\%$ in the moist layer and a linearly decreasing relative humidity above the moist layer with smooth transitions between the different

layers:

$$
\text{rh} = \begin{cases}
0.45, & \text{if } z < z_{\text{cb}} \\
0.45 + 0.15 \cos\left(0.5 \cdot \frac{z_{\text{cb,t}} - z}{z_{\text{cb,t}} - z_{\text{cb}}} \cdot \pi\right)^2, & \text{if } z_{\text{cb}} \leq z < z_{\text{cb}} + 500\,\text{m} \\
0.7, & \text{if } z_{\text{cb}} + 500\,\text{m} \leq z < z_{\text{ct,t}} - 500\,\text{m} \\
0.35 + 0.25 \cos\left(0.5 \cdot \frac{z_{\text{ct,t}} - z}{z_{\text{ct}} - z_{\text{ct,t}}} \cdot \pi\right)^2, & \text{if } z_{\text{ct}} - 500\,\text{m} \leq z < z_{\text{ct}} \\
0.35 - 4 \cdot 10^{-5}(z - z_{\text{ct}}), & \text{if } z \geq z_{\text{ct}}
\end{cases}
\tag{2}
$$

The initial profiles are based on the ICE-L case. However, we omit the vertical tilt of the orographic wave as well as the vertical gradient in maximum vertical velocity. Example cross-sections from the KiD simulations are shown in Fig. 9.

Cloud microphysics are described by the CASIM module (section 2.4) as in the UM simulations. As in the UM simulations, the sensitivity to the heterogeneous freezing parameterisations as well as assumptions for the CCN activation of INP is tested as detailed in section 2.4. Together with the different settings for dynamic and thermodynamic conditions, we have a total of 45360 two-dimensional, idealised simulations.

## 2.4 The CASIM module

The Cloud-AeroSol Interacting Microphysics (CASIM) module is a recently developed double-moment cloud microphysics scheme for the UM (Shipway and Hill, 2012; Hill et al., 2015; Stevens et al., 2018; Miltenberger et al., 2018). Hydrometeors are represented by five different species, the size distribution of which is assumed to be a generalised gamma distribution with a fixed width. Hydrometeor mass and number of each hydrometeor species are computed prognostically. CASIM also includes prognostic mass and number of three soluble and one insoluble aerosol modes, for which log-normal distribution with a fixed width are assumed. Additional tracers for aerosols incorporated into hydrometeors are available, which are transported in accordance with the hydrometeors, i.e. including sedimentation. The in-cloud aerosol tracers allow for an explicit representation of immersion freezing and to investigate the vertical transport of aerosol by hydrometeor sedimentation.

Key microphysical processes to be investigated in the mixed-phase clouds are activation of aerosols to cloud droplets, heterogeneous freezing, growth (sublimation) of ice crystals by vapour deposition, aggregation of ice crystals, and sedimentation of ice phase hydrometeors. All of these processes are represented in the CASIM module. Activation of aerosol to cloud droplets is described with the parameterisation of Abdul-Razzak and Ghan (2000). For the activation of the "insoluble" aerosol category we assume a soluble fraction on the dust particles, which is prescribed as $0.01\,\%$, $0.1\,\%$ and $99\,\%$ in three sets of sensitivity simulations. The chemical analysis of measured INP by Pratt et al. (2010) suggest that a substantial soluble fraction on INPs is realistic for the considered case. The activated INP particles are then used to predict the ice crystal number concentration using parameterisations of immersion freezing from DeMott et al. (2010) (DM10), Niemand et al. (2012) (N12), Atkinson et al. (2013) (A13), Tobo et al. (2013) (T13) and DeMott et al. (2015) (DM15). For the A13 parameterisation, we assume that $25\,\%$ of the dust surface is feldspar. Deposition and contact freezing are currently not represented in CASIM, but previous studies suggest these are not of major importance for mixed-phase orographic clouds. In addition, we have conducted

simulations, in which the insoluble aerosol number concentration is directly used in these parameterisations irrespective of whether is was incorporated into liquid first. The latter is the standard approach in all models, that do not track aerosol in hydrometeors. As the observed wave-cloud reaches temperatures colder than $-38\,°\mathrm{C}$ also homogeneous freezing is important.

Homogeneous freezing of cloud droplets is parameterised in CASIM following Jeffery and Austin (1997). In order to test the impact of homogeneous freezing on the simulated cloud microphysical structure and in particular the ice crystal number concentration, an additional simulation has been conducted, in which homogeneous freezing is switched off ("nohom", heterogeneous freezing according to DM10). Thus in total, we have 21 sensitivity experiments with different representation of immersion freezing. For the sedimentation of ice phase hydrometeors we use fixed diameter-fallspeed relations. For ice crystals the mass $\mathrm{m_i}$ is related to the mean particle diameter $\mathrm{D_i}$ via $\mathrm{m_i} = \frac{\pi}{6} \cdot 200\,\mathrm{kg\,m^{-3}}\mathrm{D_i^3}$. The fallspeed $\mathrm{v_i}$ is then computed according to $\mathrm{v_i} = 71.34\,\mathrm{m^{0.3365}\,s^{-1}} \cdot \mathrm{D_i^{0.6635}}(\rho_0\rho^{-1})^{0.5}$, where $\rho$ is the air density. The sedimentation fluxes will be sensitive to the parameters used in the mass-diameter and diameter-fallspeed relations, but we leave exploring this sensitivity to a future study.

## 2.5 Trajectory analysis

Kinematic air mass trajectories are computed to detect changes in specific humidity and aerosol number density due to sedimenting hydrometeors in the wave cloud. Trajectories are calculated with the Lagrangian Analysis Tool (Sprenger and Wernli, 2015), which has been adapted to UM output, from the resolved wind-field at $5\,\mathrm{min}$ temporal resolution. For the KiD model, trajectories are calculated analytically based on the prescribed wind field (eq. 1).

## 3 Comparison of modelled cloud properties to observational data

On the $16^{\mathrm{th}}$ November 2007 a wave cloud forming in the lee of the Medicine Bow National Forest of Wyoming was observed with three subsequent aircraft passes through the cloud at different altitudes. All flight legs are along or against the average wind direction. The average temperature of the three flight legs is $-25\,°\mathrm{C}$ (leg A, $\mathrm{z} \approx 6.9\,\mathrm{km}$, $\sim 2040$ UTC), $-27.5\,°\mathrm{C}$ (leg B, $\mathrm{z} \approx 7.2\,\mathrm{km}$, $\sim 2100$ UTC) and $-31\,°\mathrm{C}$ (leg C, $\mathrm{z} \approx 7.7\,\mathrm{km}$, $\sim 2120$ UTC). The cloud had an along-flow extension of about $40\,\mathrm{km}$ and a vertical extension of at least $1\,\mathrm{km}$. In the UM simulations a wave cloud of similar extent appears at the same location and roughly the same time ($\pm 20\,\mathrm{min}$). A horizontal cross-section of the modelled cloud at $\sim 7.2\,\mathrm{km}$, i.e. the mean altitude of flight leg B, is shown in Fig. 1 a together with the flight tracks. The modelled vertical cloud structure at $42.05\,°\mathrm{N}$ is shown in Fig. 1 b together with a projection of the aircraft legs on the plane of the cross-section. These plots already indicate that modelled cloud location and extent agree well with the observed cloud. In the remainder of this section we compare the observed and modelled cloud microphysical structure in more detail.

## 3.1 Thermodynamic conditions

The geometry of wave clouds is strongly controlled by the upstream humidity and temperature profile as well as the vertical velocity field.

Fig. 2 a shows a comparison of the upstream temperature profile. The air temperature in the model is slightly higher than observed at all vertical levels, if evaluated at the time and location of the aircraft observations, with a deviation of about 2 K for flight leg A and less than 0.1 K for flight leg C. The model suggests that the upstream temperature varied by up to 2.5 K during the time window of the observations, i.e. between 2040 UTC and 2120 UTC.

The upstream specific humidity is compared in Fig. 2 b. In general the model is somewhat more humid than observed at the time and location of flight leg C with a deviation of about $0.2\,\mathrm{g\,kg^{-1}}$. The model also suggests a quite large variability of the upstream specific humidity (roughly by a factor of 2) in the time window of the observations with a gradual moistening before 2100 UTC and a subsequent drying. As all observation data are within the modelled spread of specific humidity values, the agreement is fairly good. As the temporal evolution or zonal variation of the humidity profile is not well characterised by the observations, it is not straightforward to assess if and to what degree the differences between model and observed specific profiles impact the condensate content along the flightpaths.

In Fig. 3 the observed vertical velocity along the three flight legs is compared to the modelled vertical velocity. While in the figure we also show the vertical velocity interpolated onto the flight path (dark blue), for the analysis we use hypothetical flight paths, which are parallel to the mean modelled streamline (grey lines). Hypothetical flight paths have a horizontal spacing of 250 m in zonal direction and run through the centre of the wave clouds, i.e. have a peak vertical velocity larger than $2.5\,\mathrm{m\,s^{-1}}$. Using these hypothetical flight paths instead of the actual aircraft track eliminates the impact of slightly different horizontal wind direction in model simulations and the observed flow. The mean flow is from west to east, i.e. from left to right in these plots, and the cloud forms at the first peak in vertical velocity. The amplitude of the wave in terms of the vertical velocity is well captured in the model at all three altitudes with maximum deviations of less than $1\,\mathrm{m\,s^{-1}}$. Note that the uncertainty of the vertical velocity observations can be up to several tenth of $1\,\mathrm{m\,s^{-1}}$ (e.g. Field et al., 2012). The width of the positive vertical velocity peak is slightly larger in the model than in the observations and the peak occurs slightly further east. The secondary peaks in vertical velocity downstream of the main wave are less well captured, particularly at flight leg C (Fig. 3 a). For the cloud formation, the vertical displacement of air parcels is more important than the maximum vertical velocity. The vertical displacement depends on the amplitude, wavelength, and vertical structure of the wave. As vertical velocity observations are only available along the flight track, it is not possible to rigorously evaluate the modelled vertical displacement.

In summary, the modelled air temperature deviates less than 1 K from observations, the specific humidity less than $0.2\,\mathrm{g\,kg^{-1}}$, and the vertical velocity less than $1\,\mathrm{m\,s^{-1}}$. To our knowledge this is the first study, in which a direct comparison of aircraft measurements and simulations from a regional numerical weather prediction is done. There are many source of uncertainty in regional numerical weather prediction models including uncertainty in the analysis used for initial and boundary conditions, the representation of orography, drag, dynamics, and microphysics. In addition, upstream conditions vary in time, which is not fully captured by the aircraft measurements. Given these issues the agreement between modelled thermodynamic and dynamic conditions seems to be sufficiently well for for an in-depth comparison of the cloud microphysical structure as well as investigations of the vertical fluxes of water vapour and aerosol. Due to the small temperature bias in the model, in the following we always compare the aircraft data with the model data 200 m above the altitude of the flight track. This eliminates the temperature bias (SI Fig. 1) and allows for a better comparison of the ice nucleation.

## 3.2 Cloud structure

The microphysical data along the various aircraft legs allow for a detailed analysis of the microphysical processes due to the
280 mainly laminar flow in the wave clouds, albeit not providing a truly Lagrangian perspective. Note, that all flight legs are along
or against the average wind direction, i.e. streamlines are crossed at least twice (see also Field et al. (2012)). The in-cloud,
updraft dominated region of the flight legs is characterised by a relatively constant air temperature (variations $< 0.5$ K) and
specific humidity (variations $< 0.1$ g kg$^{-1}$) (Fig. 4 a-c, SI Fig. 1) in both the model and the observational data. The constant
specific humidity reflects water saturated conditions given the observed constant in-cloud temperature. Consistent with the
285 similar temperature in model and observation, the in-cloud specific humidity is very similar in both data-sets. This is partly
by design as the measured specific humidity was corrected such that the relative humidity is on average $100\%$ in regions with
liquid water content larger than $0.02$ g kg$^{-1}$ (Heymsfield et al., 2011).

The deviations in the spatial distribution and amount of total condensate content between model and observations are larger than
in all other variables considered so far (Fig. 4 d-f). In the upstream, updraft dominated cloud section, i.e. west of $\sim -105.1\,^{\circ}$ E,
the total condensate amount is clearly larger than in the observations for flight leg A and C. For these flight legs total condensate
data from the various available sensors agrees well in this part of the cloud (deviations of maximum value less than $3\%$ and
$20\%$, respectively). For flight leg B, King liquid water probe measured about twice the amount of condensate than the CVI.
The King liquid water probe data agrees with the model data within $60\%$ ($10\%$, $100\%$) for flight legs A (B, C) for the peak
value. In most model runs as well as in the observational data there is little ice in this part of the cloud (Fig. 5) and hence
the total condensate is controlled by the upstream humidity and the total lifting up to the considered point. Given the small
deviations in upstream humidity between model and observations, the higher modelled total condensate values are likely due to
the somewhat larger vertical velocities, which together with the similar horizontal wavelength result in larger vertical displace-
ments of air parcels than in the observations (Fig. 3). Simulations using the A13 parameterisation have an even higher total
condensate amount. In these simulations glaciation occurs very early (Fig. 5), hence the saturation pressure over ice is relevant
for the equilibrium condensate amount and not the saturation pressure over water. In the cloudy region further downstream, i.e.
downstream of $\sim -105.1\,^{\circ}$ E, observations indicate a large increase in condensate amount, despite the prevailing downdraft.
Note observational data from various sensors diverge in this part of the cloud. In the model, there is a small increase of total
condensate downstream of $\sim -105.1\,^{\circ}$ E most conspicuous for flight leg C (Fig. 4 f). This increase is, however, smaller than
in the observations, in particular for flight legs B and A. In the simulation without homogeneous freezing, the increase is absent
suggesting that the increase in condensate is due to homogeneously formed ice crystal being transported in the downdraft. It is
likely that the increase observed is due to the same mechanism.

The the horizontal extent of the cloud, in which liquid hydrometeors are present, is similar in the observations and the model
simulation further supporting the above conclusion of a good representation of the thermodynamic structure of the wave cloud
in the model simulations (SI Fig. 2). As discussed for the total condensate, the liquid water content is overestimated by the
310 model most likely due to differences in the vertical displacement or upstream humidity of the air parcels. The cloud droplet
number concentration deviates by less than $20$ cm$^{-3}$ between model and observations for all simulations except those using

A13, for which cloud droplets are depleted due to very efficient heterogeneous freezing (SI Fig. 2).

The comparison of frozen hydrometeor mass mixing ratios shows that the modelled onset of significant cloud glaciation is roughly consistent with the observations along the flight leg B and C at $\sim -105.1\,°$E, but occurs later on the flight leg A (model: $\sim -105.13\,°$E, observations: $\sim -105.18\,°$E; Fig. 5 a-c). The steep increase in the mass mixing ratio on flight legs B and C (downstream of $\sim -105.1\,°$E) is associated with a rapid increase in ice number concentration in the model (Fig. 5 d, e) and occurs in the downdraft region. While both the 2D-C and the 2D-S data agree quite well in terms of the ice crystal number concentrations (within factor 2), the estimated ice crystal mass diverges. The large increase of ice crystal mass and number in the downdraft region is likely due to the downward transport of ice crystals formed by homogeneous freezing by the descending air with a minor contribution of sedimentation. The importance of homogeneously formed ice in the downdraft region is supported by the divergence of ice crystal number concentrations in the simulations with and without homogeneous freezing (compare orange and blue line in Fig. 5 d-f). In the observations ice mass and number concentration increase also in the downdraft region. However, the ice water content, at least in the 2D-C data, increases before the strong increase in ice crystal number concentration. In the model the steep increase in crystal number concentrations occurs earlier than in the observations. We hypothesis that the earlier increase of ice water content, which generally coincides with the start of the downdraft regions is due to larger displacements of air parcels in the model (somewhat larger vertical velocities, Fig. 3) and hence a stronger downward transport of homogeneously formed ice crystals. Alternatively, a too early onset of homogeneous freezing or a too rapid sedimentation of ice crystals could also lead to the observed differences between model and observation. Based on the available data, none of these options can be ruled out. The first ice crystals larger than $50\,\mu$m appear in approximately the same location as in the observations at all altitudes, however with much larger concentrations. This suggests a too large droplet mass in the freezing event consistent with the overestimation of liquid condensate. Maximum ice crystal concentrations for most simulations agree within a factor 2 on flight legs A and B, while they are about a factor 10 larger on the flight leg C (Fig. 5 d-f). However, if not only the maximum concentration is considered, modelled and observed ice crystal number concentration is within a factor 2 only for the lowest flight level with differences of almost an order of magnitude on the higher flight levels. As pointed out earlier, simulations using the A13 parameterisation strongly overestimate the ice crystal number concentration inducing a too early onset of glaciation. Different assumptions on the CCN activation of dust particles (different line styles of the same colour in Fig. 5) have only a small impact on the modelled ice crystal mass and number concentrations, with the largest impact in simulations using the N12 parameterisation and flight leg C. Even for simulations with N12 the resulting differences are much smaller than the difference to the observed time series and it is not clear whether representing CCN activation of dust particles yields an improvement based on these. As expected the location, at which ice crystals first appear, is shifting slightly downstream in simulations with a smaller soluble fraction on the dust particles. The horizontal extent of the ice tail in the model is overestimated for all flight legs, except flight leg A (Fig. 5c, f). The longevity of ice crystal in the model is very likely related to the smaller average ice crystal mass, i.e. the ratio of ice crystal mass mixing ratio and number concentration, and the untuned parameters used to compute the mean fallspeed from the ice crystal diameter.

For the comparison of liquid and ice hydrometeor number concentrations and mass mixing ratios it is important to also consider limitations of the observational data. Most importantly, only data for ice crystals larger than $50\,\mu$m are used. This has been

taken into account by estimating the number and mass of ice crystals larger than $50\,\mu\mathrm{m}$ from the modelled total mass and number concentrations using the prescribed distribution and shape parameter in CASIM. If small ice crystals were abundant in the wave clouds, this would improve the match of model and observations in the cloud region dominated by heterogeneously

formed ice crystals, but deteriorate it in the region dominated by homogeneously formed crystal. Another issue is that the 2D-C may detect large drizzle, which hence would be misclassified as ice, and the CDP measurements may include small ice particles (D'Alessandro et al., 2019). It is very unlikely for drizzle drops to be present in wave clouds due to the cold temperatures and the short time parcels reside in the cloud ($<\ 30\,\mathrm{min}$) and hence no significant bias of the ice crystal number concentrations is expected. If some ice particles would be included in the CDP measurement, this would also only have a very

limited impact on the comparison as ice number concentrations are about three orders of magnitude smaller than cloud droplet number concentrations.

In summary, observations and model simulations (except those using the A13 parameterisation) agree on the overall microphysical structure of the cloud with ice particles and cloud droplets co-existing, a similar location for the appearance of first ice crystals and ice crystals from the homogeneous freezing zone affecting cloud properties in the downdraft region. Despite

the overall good agreement in the structure of the wave-cloud, modelled and observed total condensate amount as well as ice crystal number concentration deviate clearly. The former is most likely caused by an overestimation of parcel vertical displacement in the model, while the overestimation in initial ice crystal number is either related to the heterogeneous freezing parameterisations used or a too large diameter of the newly formed ice crystals. Assuming the same number of crystals being nucleated at a specific temperature, a large initial crystal mass results in a larger fraction of these ice crystals being detected

early on, as their size more quickly exceed the detection limit of $50\,\mu\mathrm{m}$. In the following section we investigate in more detail how different heterogeneous freezing parameterisations influence the spatial distribution of ice crystal number concentration.

### 3.3 Temperature dependency of heterogeneous ice formation

The main difference between the various heterogeneous freezing parameterisations is the temperature dependency of INP and the prefactors specifying the INP fraction of dust particles, as illustrated in Fig. 6. While observing the impact of INP temper-

370 ature dependence in most clouds is challenging due to impacts of sedimentation and strong vertical motion, the laminar flow and quasi-Lagrangian nature of aircraft observations in orographic clouds may facilitate observations of the signature of INP temperature dependence in ice crystal number concentrations. In order to test this hypothesis in the model we focus on the ice crystal number concentration in the updraft region of the cloud, i.e. i.e. west of $-105.1\,^\circ\mathrm{E}$, which is not influenced by homogeneously nucleated ice crystals (compare orange and blue lines in Fig 5 d-f). This modelled ice crystal number concentration

is compared with the ice crystal number concentration expected from the heterogeneous freezing parameterisation based on the temperature and the upstream dust profile (compare coloured markers and lines in Fig. 6): In general these agree very well suggesting that we can use observations of ice crystal number concentration from the updraft region of orographic wave clouds to constrain the temperature dependence of INP concentration. The observed ice crystal number concentrations from the 2D-C for the different flight legs are shown in the black box-plots in Fig. 6. These data suggest a very weak temperature dependency

of heterogeneous freezing in the observed wave cloud, which is only consistent with the DM10 parameterisation. All other

parameterisations appear to have a too strong temperature dependence. However, the temporal evolution of the upstream dust concentrations has to be considered as this can result in shallower or steeper temperature apparent temperature dependence. As the upstream dust profile was not monitored continuously and is constructed from the upstream observations along the flight legs, only the potential impact of time-varying dust concentrations can be assessed. For this we use the minimum and maximum observed upstream dust concentrations, irrespective of the observed altitude, to derive the resulting impact on the expected ice crystal number concentration: The shaded area in Fig. 6 indicates the spread in expected number concentrations, while the dashed lines represent a scenario with continuously decreasing upstream dust concentrations. If the latter scenario is considered, the observations are consistent also with the T13 simulation. Reliable observational data of the ice crystal size distribution are only available for particles larger than $50\,\mu$m. Hence, the analysis here considers only the largest observed ice crystal number concentration in the updraft region. While this limits the impact of different mean droplet volumes during freezing and potential differences in depositional growth, it introduces additional uncertainty into the comparison.

The comparison shows that all heterogeneous freezing parameterisation, except that from Atkinson et al. (2013) are compatible with the observations within the anticipated uncertainty range. For simulations with the A13 parameterisation we assume feldspar to be constitute $25\,\%$ of the dust surface, which is at the upper end of the composition of natural dust Atkinson et al. (2013). If a value closer to the lower bound of $1\,\%$ would have been used, A13 would be closer to the other parameterisations at $-25\,^\circ$C but still predicts too high ice crystal concentrations at colder temperatures (SI Fig. 3). The experiment closest to the observations is DeMott et al. (2010), followed by DeMott et al. (2015) and Tobo et al. (2013) (compared black box-plots with dashed lines and shading in Fig. 6). However, it is unclear whether these parameterisations are most applicable in other cases or other geographic regions, as the INP activity is known to strongly depend on the chemical composition and size distribution of aerosols (e.g. Petters and Wright, 2015). Nevertheless, the presented results suggest that wave clouds can be used as natural laboratories to investigate the temperature dependence of heterogeneous freezing. To formulate constraints on the parameterisations observations from more wave cloud events are necessary. In any future campaigns targeting orographic wave clouds, an emphasis should be placed on characterising the full ice crystal size distribution as well as the temporal (and spatial) variation of the upstream aerosol concentration.

## 4 Modification of water vapour and aerosol profiles

### 4.1 ICE-L case

One important impact of wave clouds on the evolution of the larger-scale atmospheric state is modification of water vapour and aerosol profiles through sedimentation of hydrometeors (in addition to the alteration of radiative fluxes). Vertical transport of water vapour and aerosols occurs in all clouds, but it is likely easier to observe these fluxes in wave clouds. The downward transport of water vapour depends strongly on the size and fall velocity of the formed hydrometeors. Thick warm-phase orographic clouds are known to produce significant precipitation. The downward water vapour transport is much smaller for mixed-phase wave clouds due to the smaller size and fallspeeds of ice crystals. However, according to the model simulations

the largest ice crystal diameters are on the order of $400 - 600\,\mu\mathrm{m}$ (not shown), which results in vertical displacement of about $700\,\mathrm{m}$ due to sedimentation over the roughly $30\,\mathrm{min}$ air parcels spend inside the cloud. Here, we quantify the downward transport of water and aerosol by considering the change in total water or aerosol number concentrations along trajectories through the wave cloud ($\Delta\mathrm{q_t}(\mathrm{z_0})$). We refer to the vertically integrated increase of total water in the lower part of the profile (equal to the decrease in total water in the upper part of the profile) as the total downward moisture transport ($\Delta\mathrm{q_t}$). Fig. 7 a and b show the Lagrangian change in total water along backwards trajectories starting in the lee of the cloud for simulations with the A13 and DM10 heterogeneous freezing parameterisations. In the time period after $80$ min a typical sedimentation signal is obtained with a depletion of the total water content in parcels above about $7\,\mathrm{km}$ and an increase in parcels below. At earlier times, this pattern is repeated twice in the vertical and a closer inspect reveals that there are two cloudy layers in the UM simulations, one formed by homogeneous nucleation and the lower one by heterogeneous freezing. To assess the differences between simulations with different heterogeneous freezing, average profiles for the two time periods from all simulations are shown in Fig. 7 c and d. The shading indicates the variability resulting from different starting latitudes of the trajectories. The mean profiles for simulations with all parameterisations except A13 are very similar and much smaller than the temporal variability. The larger INP concentrations predicted by A13 lead to much larger change in total water content $\Delta\mathrm{q_t}(\mathrm{z_0})$ particularly in the first part of the considered time period. The assumptions on the CCN activation of dust lead to very small differences in $\Delta\mathrm{q_t}(\mathrm{z_0})$ (SI Fig. 4 a, b).

The CASIM microphysics explicitly considers the vertical transport of dust particles by hydrometeor sedimentation and therefore allows us to quantify the downward transport of aerosol by the wave cloud. The Lagrangian change of aerosol content is shown in Fig. 8. The vertical structure is different to $\Delta\mathrm{q_t}(\mathrm{z_0})$, with aerosol depletion only occurring at the very top of the cloud (above $\sim 9.7\,\mathrm{km}$) and increases in aerosol number concentrations mainly towards cloud base. The modifications of the dust profiles are more sensitive to changes in the heterogeneous freezing parameterisation than those of the total water content, with larger changes also in the shape of the profiles. However, the differences are again smaller than the temporal variability. The treatment of the CCN activation of dust (using all dust for heterogeneous freezing or presenting activation assuming some soluble fraction on dust particles) has a much larger impact on the vertical aerosol transport than on the moisture transport. The resulting differences in the profile are on the same order of magnitude as the temporal variability (SI Fig. 4 c, d).

It would be interesting to constrain the downward transport with observational data, in particular given the uncertainties surrounding diameter-fallspeed relations often used in bulk models. The maximum change in $\mathrm{q_t}$ of about $0.1\,\mathrm{g\,kg^{-1}}$ is, however, smaller than the temporal variation of the specific humidity (Fig. 2) during the average time a parcel needs to transit through the wave cloud (i.e. $\sim 30$ min). As the aircraft data do not provide information on the temporal evolution of upstream humidity, it is not possible to use the aircraft data to constrain the vertical moisture transport by sedimentation. In addition, for such an assessment the construction of air parcel trajectories from the observed velocity field would be required. While this is in principle possible (e.g. Field et al., 2012), for the assessment of downward moisture transport the error in the upstream positions of air parcels would need to be smaller than $500\,\mathrm{m}$ owing to the vertical gradient of upstream specific humidity. This is not feasible given the sparse observations of velocity (only sampled along flight legs) and the uncertainty in measured vertical velocity. However, detailed observations of the 3D velocity field for example with an on-board Lidar system and a

better characterisation of the upstream and downstream humidity profiles, e.g. sampling in a quasi-Lagrangian manner, there
is a potential for future field campaigns to constrain vertical transport of moisture by sedimenting hydrometeors from wave
clouds.

Because wave clouds offer such an opportunity to detect sedimentation mediated vertical transport of moisture and aerosol we
assess in the following section how its amplitude depends on the upstream thermodynamic conditions, which determine the
cloud thickness and cloud top temperature, and on the horizontal wavelength of the gravity wave, that controls the horizontal
extent of the cloud.

### 4.2 Downward moisture transport by sedimentation in idealised simulations

The modification of moisture and aerosol profiles by hydrometeor sedimentation is investigated for the ICE-L case study in the
previous section. However, the cloud-integrated sedimentation fluxes will vary for different wavelength, cloud top temperatures
and cloud thicknesses and so will their impact on the vertical profiles of aerosol and moisture. To assess these dependencies,
we use two-dimensional, idealised simulations with the KiD-model (section 2.3). Using an idealised model for this assessment
allows us to vary the wavelength of the gravity wave, which would require changing the topography in the Unified Model.
In addition, we can carry out a large number of simulations sampling a large proportion of the relevant phase-space, which
would not be possible with the UM due to the much larger computational costs. But we are able to link back to the case study
by including the observed case in the phase space explored. Two exemplary realisations of wave cloud in the KiD-model are
shown in Fig. 9 along with the profiles of Lagrangian changes in moisture ($\Delta_{\mathrm{Lagr}}q_t$) and aerosol $\Delta_{\mathrm{Lagr}}m_{du}$. As in the UM
simulations, the profiles of moisture and aerosol changes have distinctly different shapes: While aerosol changes are concentrated at cloud top and cloud base, moisture changes occurring throughout the cloud with peak values in the upper and lower
half of the cloud, respectively.

To explore the variation of the downward transport as a function of cloud geometry, we focus on the cloud-scale downward
moisture (aerosol) transport $\Delta q_t$ ($\Delta m_{du}$), which we define as the integral of positive $\Delta_{\mathrm{Lagr}}q_t$ ($\Delta_{\mathrm{Lagr}}m_{du}$). Note that the integral over negative $\Delta_{\mathrm{Lagr}}q_t$ ($\Delta_{\mathrm{Lagr}}m_{du}$) gives the same results due to mass conservation, albeit of course with a different sign
(not shown). For further analysis we split the sedimentation flux into sedimentation of liquid ($\Delta_{\mathrm{Lagr}}q_{t,l}$) and frozen hydrometeors ($\Delta_{\mathrm{Lagr}}q_{t,l}$), which display a different dependence on the explored phase-space control parameters. Fig. 10 a summaries
$\Delta q_{t,f}$ for all investigated wave periods (abscissa) and cloud top temperatures (ordinate) for a cloud depth of $2000\,\mathrm{m}$. SI Fig. 5 a
is the equivalent for $\Delta_{\mathrm{Lagr}}q_{t,l}$. $\Delta_{\mathrm{Lagr}}q_{t,l}$ is only important for cloud top temperatures warmer than $\sim -30\,^{\circ}\mathrm{C}$ (SI Fig. 5 b).
In both the UM and the KiD model rain formation is included as is the sedimentation of cloud droplets and rain drops. Rain
formation is found in all simulations to be negligible, with the rain mass mixing ratio at least one order of magnitude smaller
than the mass mixing ratio of any other hydrometeor. This is due to the short in-cloud residence timescales ($< 30\,\mathrm{min}$), which
according to the timescale analysis in Stevens and Seifert (2008) and Miltenberger et al. (2015) is too short for significant
rain formation. As the dependence of $\Delta_{\mathrm{Lagr}}q_{t,l}$ is quite different from $\Delta_{\mathrm{Lagr}}q_{t,f}$ and considerations on $\Delta_{\mathrm{Lagr}}q_{t,l}$ are already
published in Miltenberger et al. (2015), the following analysis will predominantly focus on $\Delta_{\mathrm{Lagr}}q_{t,f}$. $\Delta_{\mathrm{Lagr}}q_{t,f}$ has also been
computed from the UM simulations and is shown by the colour-filled circle at $T = 1800\,\mathrm{s}$ and $t_{ct} = -45\,^{\circ}\mathrm{C}$, which corre-

sponds to the average cloud top temperature and residence time of parcels in the orographic cloud for the ICE-L case. In the UM only the central section of the wave-cloud with largest vertical velocities is considered. $\Delta q_{t,f}$ for the three-dimensional

UM simulation and the idealised KiD simulation are comparable in value. This justifies the use of the KiD model to explore the dependence of $\Delta q_{t,f}$ on the upstream thermodynamic profile and the wave period.

The most prominent feature in the variation of $\Delta q_{t,f}$ over the sampled part of the phase-space is the strong increase of $\Delta_{\mathrm{Lagr}} q_{t,f}$ at about a cloud top temperature of $-37\,^\circ\mathrm{C}$, which is due to the onset of homogeneous freezing and hence a large increase in the frozen water content available for sedimentation. For all cloud top temperatures $\Delta_{\mathrm{Lagr}} q_{t,f}$ increases towards longer wave

periods as expected. These general patterns are consistent for all cloud thicknesses investigated (not shown). The downward moisture transport increase with larger cloud thickness, but the impact of cloud thickness is smaller than that of wave time period and cloud top temperature (not shown). Hence, the discussion in the following focusses on a single cloud thickness, although all sensitivity experiments are included in the formulation of the conceptual model. Consistent with the UM simulations, the parameterisation used for heterogeneous freezing impacts the downward moisture transport: Fig. 10 b shows the

maximum difference between any two simulations with the same wave period, cloud top temperature and cloud thickness, but different heterogeneous freezing parameterisations (20 simulations for each combination of wave period, cloud top temperature and cloud thickness). For the UM simulation the variability is about a factor 5 larger (colour-filled circle in Fig. 10 b), which is mainly due to the low values for the simulation with DM10 and $\epsilon = 0.01$. The impact of the parameterisation choice is largest for cloud top temperatures just below the onset of homogeneous freezing (see e.g. Fig. 6). In this part of the parameter space

$\Delta_{\mathrm{Lagr}} q_t$ varies by up to a factor 10 between simulations with different heterogeneous freezing parameterisations. Differences between simulations with different heterogeneous freezing parameterisations are largest for wave periods larger than $800\,\mathrm{s}$ and cloud top temperatures between $\sim -30\,^\circ\mathrm{C}$ and $\sim -38\,^\circ\mathrm{C}$.

The downward transport of aerosol $\Delta_{\mathrm{Lagr}} m_{\mathrm{du}}$ is summarised in Fig. 10 c and d. $\Delta_{\mathrm{Lagr}} m_{\mathrm{du}}$ and its variation with cloud microphysical parameterisation choices is again very similar to the values obtained from the UM simulation (colour-filled circles in

Fig. 10 c, d). The aerosol downward transport increases, similar to the downward moisture transport, with longer wave periods and towards colder cloud top temperatures. The increase with decreasing cloud top temperature is, however, smoother than for $\Delta_{\mathrm{Lagr}} q_t$. Towards the onset of homogeneous freezing most heterogeneous freezing parameterisations predict that a substantial fraction of dust is activated as INP and hence there is no step-change at the onset of homogeneous freezing. Differences between simulations with different settings in the cloud microphysics are largest for wave periods larger than $600\,\mathrm{s}$ and cloud top

temperatures between $\sim -19\,^\circ\mathrm{C}$ and $\sim -28\,^\circ\mathrm{C}$.

A conceptual model of the moisture transport by sedimenting frozen hydrometeors provides insight into the key variables controlling the modification of the moisture profile and may be used to represent these in models with a lower spatial resolution. Similar to previously proposed conceptional models for orographic precipitation Smith (1979); Smith and Barstad (2004); Seifert and Zängl (2010); Miltenberger et al. (2015), we chose an ansatz based on the consideration of the characteristic

timescales of the cloud:

$$\Delta q_{t,f} = \left( \left( G_{\mathrm{pot}} - G_{\mathrm{nuc}} \right) \left( 1 - \exp\left( -\frac{\tau_{\mathrm{ic}}}{\tau_{\mathrm{dep}}} \right) \right) + G_{\mathrm{nuc}} \right) \cdot \left( 1 - \exp\left( -\frac{\tau_{\mathrm{ic}}}{\tau_{\mathrm{sedi}}} \right) \right) \tag{3}$$

The first term on the left side of the equation describes how much water is transferred from the gas-phase to frozen condensate due to depositional growth and freezing, while the second term describes the sedimentation of the condensate. The key variables are (i) the potential condensate $G_{pot}$, which is the maximum cloud condensate possible given thermodynamic constraints, initial humidity and vertical displacement, (ii) the in-cloud residence time $\tau_{ic}$, i.e. the time available for cloud microphysical processes, (iii) the timescale for depositional growth of ice hydrometeors $\tau_{dep}$ and (iv) the timescale for sedimentation $\tau_{sedi}$. Note that in contrast to parcel-oriented formulations these timescales refer to the entire cloud and not to individual air parcels. Finally, $G_{nuc}$ denotes the condensate formed during ice crystal nucleation via homogeneous or heterogeneous freezing. A similar approach has been suggested by Seifert and Zängl (2010) and Miltenberger et al. (2015) for describing the precipitation formation in warm-phase orographic clouds. As we show in the following, all parameters in equation 3 can be estimated from the upstream thermodynamic profiles and expected vertical displacement.

As mentioned above, we focus here on the sedimentation flux of frozen hydrometeors. For cloud top temperatures warmer than about $\sim -30\,^\circ\text{C}$, the impact of cloud droplet sedimentation is comparable or larger to that of frozen hydrometeor sedimentation (SI Fig 5 b). In contrast to $\Delta_{Lagr}q_{t,f}$, $\Delta_{Lagr}q_{t,l}$ depends in our set of experiments only on $\tau_{ic}$. The main reason for this is that a saturation adjustment scheme is used in the UM and KiD model and only a specific maximum vertical displacement is considered (see section 2.3). Hence, for the following ansatz is chosen for moisture transport by sedimenting cloud droplets:

$$\Delta q_{t,f} = G_{pot,l}\left(1 - \exp\left(-\frac{\tau_{ic}}{\tau_{sedi,l}}\right)\right) \tag{4}$$

The potential liquid condensate $G_{pot,l}$, i.e. the difference between the upstream specific humidity and the saturation water content over water at the coldest point along the trajectory, and $tau_{ic,l}$ can be estimated using the same procedure as outlined below for $G_{pot}$ and $\tau_{ic}$ by considering the saturation mass mixing ratio over water instead of that over ice. The sedimentation timescale $\tau_{sedi,l}$ can be estimated by using the profile of $G_{pot,l}(z_0)$ together with the typical cloud droplet number concentration (here $70\,\text{cm}^{-3}$), the fallspeed-diameter and mass-diameter relationships used in the model and the cloud depth. Note that despite being significant compared to the frozen hydrometeor sedimentation flux, fluxes are generally very small for temperatures warmer than $-30\,^\circ\text{C}$. In the following, we discuss in detail the estimates for variables in equation 3, in-line with the focus of the paper.

The potential condensate is the maximum condensate amount that would occur along a wave cloud trajectory if the air parcel's ice water content were in thermodynamic equilibrium, i.e. roughly the difference between the upstream vapour content and the saturation water content over ice at the coldest point along the trajectory. In warm-phase clouds the condensate amount in absence of sedimentation is often close to the potential condensate as a result of fairly small vapour deposition timescales ($\sim 1\,\text{s}$), as e.g. used in saturation adjustment parameterisations. However, in mixed- and ice-phase clouds the potential condensate is typically not realised due to the longer timescales for depositional growth (in the order of $1000\,\text{s}$). $G_{pot}$ is not used as a measure of the condensate formed in the cloud, but as a "virtual" reservoir species from which condensate can be formed. Along air parcel trajectories $G_{pot}$ can be directly computed as the difference between the upstream specific humidity and the saturation pressure over ice at the coldest point along the trajectory, if latent heating from phase-changes of water are neglected. Using trajectory data from the KiD experiments, the variation of $G_{pot,Lagr}$ with the wave period and cloud top temperature can

be quantified (Fig. 11 a). Further $G_{pot}$ can be computed from the wave amplitude A and the upstream temperature $t_0$, specific humidity $q_{v,0}$ and pressure $p_0$ profiles by assuming dry-adiabatic ascent of the parcel (lapse rate $\gamma$) and a hydrostatic balanced atmosphere:

$$G_{pot}(z_0) = (q_{v,0} - q_{i,sat}(t_0 + A\gamma))\, p_{z_0+A} \qquad (5)$$

Integrating above equation over all altitudes where $q_{v,0} > q_{i,sat}(t_0 + A\gamma)$ gives an estimate of $G_{pot}$, which for our KiD simulations deviates less than $5\%$ from the Lagrangian estimate shown in Fig. 11 a (SI Fig. 6 a).

Another important cloud microphysical variable, that will be required for parameterising the characteristic timescales is the number of ice crystals in each cloud. To characterise the variability across the different clouds, we use only the maximum possible number of ice crystals $n_{i,\,max}$ formed either by homogeneous or heterogeneous freezing. In the Lagrangian data, this

is the integral of homogeneous and heterogeneous nucleation rates along the trajectory passing just below cloud top. Fig. 11 b shows that $n_{i,\,max,Lagr}$ depends strongly on cloud top temperature with a major increase around $t_{ct} \approx -38\,°C$ reflecting the transition to clouds dominated by homogeneous freezing. For clouds with colder cloud tops there is also a clear dependence on the time period of the wave clouds, reflecting the interaction between the nucleation and growth of newly formed ice crystals (e.g. Kärcher et al., 2006). For the conceptual model, we find that using the heterogeneous parameterisation used in

the KiD model together with the minimum temperature expected from the maximum vertical displacement gives a reasonable estimate for temperatures warmer than $-38\,°C$. For colder cloud-top temperatures, we use the homogeneous nucleation rate from the DM10 parameterisation (consistent with CASIM microphysics) and a correction factor depending on the wave period: $0.932 \cdot \log_{10}(T) + 0.228$ for $t_{ct} < -42.5\,°C$ and $1.48 \cdot \log_{10}(T) - 1.48$ for $t_{ct} > -42.5\,°C$. Closely related to the ice crystal number is also the term $G_{nuc}$ describing the ice crystal mass formed by homogeneous or heterogeneous freezing.

$G_{nuc}$ can be estimated from $n_{i,max}$ and a typical particle mass $\bar{q}_i$, which can be directly obtain from the KiD simulations: $\bar{q}_i = 10^{-11.5}\,\mathrm{kg\,kg^{-1}}$ ($10^{-9.6}\,\mathrm{kg\,kg^{-1}}$) for clouds dominated by homogeneous (heterogeneous) freezing.

The in-cloud residence time $\tau_{ic}$ describes the time available for condensate and precipitation formation (e.g. for warm clouds Miltenberger et al., 2015). Here, we define $\tau_{ic}$ as time during which air parcels are super-saturated with respect to ice. This timescale $\tau_{ic,Lagr}$ can be directly quantified from the KiD-model air mass trajectories (Fig. 12 a) or analytically calculated from

the prescribed wave flow and the upstream humidity profile:

$$\tau_{ic} = T \cdot \left(1 - \arccos(1 - 0.5 \cdot \eta_{i,sat} A \pi^{-1})\pi^{-1}\right) \qquad (6)$$

with $\eta_{i,sat}$ the vertical displacement required to reach ice saturation. The deviations between this estimate and the Lagrangian metric are less than $5\%$ (SI Fig. 6 b). From the resulting vertical profile of $\tau_{ic}$ the largest timescale is selected (only considering cloudy parcels).

The depositional timescale $\tau_{dep}$ describes the characteristic timescale for the reduction of ice supersaturation for $w = 0\,\mathrm{m\,s^{-1}}$ and an ice crystal population characterised by the number concentration $n_i$ and mean ice particle diameter $d_i$. The concept of describing depositional growth of ice crystals with a characteristic timescale $\tau_{dep}$ is frequently used in literature and cloud microphysical parameterisations (e.g. Khvorostyanov, 1995): $\tau_{dep} = (g n_i d_i c_i f)^{-1}$, where $g = 4\pi \cdot (L_{ed}^2 (K_t R_d t^2)^{-1} + R_d t (D_{vtp} e_{s,i})^{-1})^{-1}$,

$L_{ed}$ the latent heat of sublimation, $K_t$ the heat conductivity, $R_d$ the specific gas constant for dry air, $D_{vtp}$ the diffusivity of water vapour, $e_{s,i}$ the saturation vapour pressure over ice, $c_i$ the capacitance of the ice crystals, and $f$ a ventilation factor. This concept needs to be extended to a single characteristic timescale for the entire cloud. To estimate this timescale we again utilise the KiD simulations. The cloud-scale deposition timescale can be estimated from the integrated deposition $D$ and freezing rates $G_{nuc}$ as well as $\tau_{ic}$ according to: $\tau_{dep,Lagr} = \tau_{ic,Lagr} \left(\log(1 - D(G_{pot} - G_{nuc})^{-1})^{-1}\right)^{-1}$. The resulting estimates are shown in Fig. 12 b. Immediately obvious is an inverse relation to the ice crystal number concentration, as expected from air parcel considerations, but this is not the sole determinant. In order to estimate $\tau_{dep}$ from the a-priori known parameters, i.e. upstream profiles and vertical displacement, we determined the following least-square fits to the KiD model data (SI Fig. 6 c):

$$
\tau_{dep} = \begin{cases}
5.29 \cdot 10^{10} \cdot n_i^{-1.94} \cdot T^{-0.558} \cdot z_c^{0.539}, & \text{if } t_{ct} \geq -34.25\,°\text{C} \\
1.71 \cdot 10^{7} \cdot n_i^{-0.764} \cdot T^{-0.716} \cdot z_c^{0.696} \cdot (n_i T)^{0.0566}, & \text{if } -44.3\,°\text{C} < t_{ct} < -34.25\,°\text{C} \\
1.34 \cdot 10^{7} \cdot n_i^{-0.749} \cdot T^{-0.271} \cdot z_c^{0.576}, & \text{if } t_{ct} \leq -44.3\,°\text{C}
\end{cases}
\tag{7}
$$

The sub-division is necessary due to the fundamentally different behaviour in the parts of the parameter space dominated by homogeneous and heterogeneous freezing, respectively.

Finally, the sedimentation timescale needs to be determined, for which we use the same approach as for the deposition timescale, i.e. diagnosing a cloud-wide timescale from the KiD model and constructing a statistical model. The timescale is estimated from the KiD model according to: $\tau_{sedi,L} = \tau_{ic,L} \left(\log(1 - \Delta q_{sedi}(D + G_{nuc})^{-1})^{-1}\right)^{-1}$. The results are shown in Fig. 12 c. The sedimentation velocity in the KiD model is described using a prescribed diameter-fallspeed relation. Consistently, the $\tau_{sedi}$ increases for clouds with larger $n_i$. In addition to this information, we find that it is necessary to incorporate information on the time period and cloud thickness in the statistical model likely due to their impact on the cloud microphysical evolution (SI Fig. 6 d):

$$
\tau_{sedi} = \begin{cases}
4.01 \cdot 10^{3} \cdot n_i^{-0.0185 \cdot T - 0.242 z_c - 0.449} \cdot T^{0.0253 t_{ct} - 0.467}, & \text{if } t_{ct} \geq -32.2\,°\text{C} \\
4.26 \cdot 10^{4} \cdot n_i^{-0.0507 \cdot T - 0.326 z_c + 1.03} \cdot T^{0.663 t_{ct} - 2.43}, & \text{if } -38.5\,°\text{C} < t_{ct} < -32.2\,°\text{C} \\
3.06 \cdot 10^{4} \cdot n_i^{0.385 \cdot T - 0.0613 z_c - 0.500} \cdot T^{-0.143 t_{ct} - 1.65}, & \text{if } t_{ct} \leq -38.5\,°\text{C}
\end{cases}
\tag{8}
$$

By using equations 3 to 8 with the described approximation of $n_{i,max}$ the total downward moisture transport by sedimentation can be computed based on the upstream dust concentration, the upstream profiles of temperature, humidity and pressure and the maximum vertical displacement. The parameterised $\Delta q_t$ is shown in Fig. 13 a. Comparing this figure with the results from the full KiD model (Fig. 10 a) shows very similar dependencies on wave period and cloud-top temperature. Note that Fig. 10 a shows the average $\Delta q_t$ from simulations with different heterogeneous freezing parameterisation, while Fig. 13 a shows data only for simulations with DM10. Hence the differences in absolute values. The absolute values from the conceptual model agree well with the simulations from the full KiD model with discrepancies mostly smaller than 30 % (Fig. 13 b, SI Fig. 7).

 **5 Conclusions**

Orographic wave clouds impact atmospheric flow by interacting with radiative fluxes and by modifying the moisture and aerosol profiles. Furthermore due to the laminar flow they are ideal natural laboratories to explore cloud microphysical processes along the wind (time) direction. Here, we compare simulations with the Unified Model (UM) including the recently developed Cloud-Aerosol Interacting Microphysics (CASIM) module to observations from the ICE-L measurement campaign, which took place in 2007 over the mountain states of the US.

High-resolution simulations with the UM capture the thermodynamic structure and vertical velocity field very well with deviations of less than $1\,\mathrm{K}$ for air temperature, $0.2\,\mathrm{g\,kg^{-1}}$ for specific humidity and $1\,\mathrm{m\,s^{-1}}$ for vertical velocity. The overall cloud microphysical structure of the cloud is similar to the observations, although there are significant difference in the impact of homogeneous freezing, the extent of ice tail of the cloud and the size distribution. Some of the differences could be explained by an overestimation of the vertical displacement in the model, but problems with the cloud microphysical parameterisation can also not be excluded. More detailed information on the 3D wind field should be considered in future studies. Several heterogeneous freezing parameterisations have been proposed in recent years and we explicitly tested their impact on the cloud structure. Most tested heterogeneous freezing parameterisations gave very similar results. The main difference between simulations with the different schemes is the vertical gradient of ice crystal number concentration in the updraft region of the cloud, all other investigated cloud properties display only a very small sensitivity. For all tested parameterisations, except Atkinson et al. (2013), the vertical gradient of the ice crystal number concentration is consistent with the observations given the uncertainty in observations and their representativity. The best agreement is obtained for simulations with DeMott et al. (2010), followed by those using DeMott et al. (2015) and Tobo et al. (2013). As CASIM explicitly models dust particles in liquid and ice hydrometeors, we also tested the impact of using also dust incorporated in liquid droplets for heterogeneous freezing and of prescribing different soluble fractions on dust aerosols. Both made only very little impact on the cloud microphysical structure. Despite the well captured thermodynamic conditions and flow dynamics, vigorous conclusion about link between ice crystal number concentration and upstream aerosol, in particular the temperature dependence of heterogeneous freezing, are difficult to arrive at. For this purpose, future campaigns need to provide a better characterisation of the upstream profiles of aerosols and their temporal evolution as well as observations of the full ice crystal size distribution (here limited to particles larger than $50\,\mu\mathrm{m}$). The advance in measurement techniques over the past years allows to meet these requirements in future field campaigns.

The simulations were further used to investigate the modification of moisture and aerosol profiles by the sedimentation of hydrometeors in the wave cloud. The latter was only possible due to the novel capabilities of the CASIM module. Lagrangian estimates suggest a different vertical structure of the aerosol and moisture changes, with those for aerosols concentrated at cloud top and cloud base. However, the fairly small changes in the profiles ($< 0.1\,\mathrm{g\,kg^{-1}}$ for moisture, $< 0.1\,\mathrm{cm^{-3}}$ for aerosol) prevent to constrain the sedimentation fluxes with observations.

Two-dimensional, idealised simulations were developed to further investigate the parameter space, with a particular focus on the dependence of the moisture and aerosol sedimentation fluxes on the cloud geometry, i.e. the wavelength, cloud top temper-

ature and cloud thickness. The simulations are confined to a specific vertical displacement of roughly $900\,\mathrm{m}$ and time periods of the wave motion between $100 - 1800\,\mathrm{s}$. From the few climatological studies available the later is roughly what is expected for isolated cap clouds or lee-wave clouds. It would be interesting to extend the analysis to different vertical displacements, i.e. larger wave amplitudes. While larger (or smaller) wave amplitudes would modify the condensate formed in the cloud, we do not expect a major impact on the timescale approach discussed above. However, some of the empirical fitting parameters may change as a result of the establishment of different size distributions. Extending the analysis to different vertical displacements is beyond the scope of the present study. The sensitivity to the heterogeneous freezing parameterisation is found to be largest for wave periods larger than $1000\,\mathrm{s}$ and cloud top temperatures between $-30\,^{\circ}\mathrm{C}$ and $-40\,^{\circ}\mathrm{C}$. The modifications of the moisture and aerosol profiles are largest for clouds with long wave periods and cloud top temperatures colder than $-40\,^{\circ}\mathrm{C}$. The Lagrangian change of water content is on the order of $0.1\,\mathrm{g\,kg}$ and that of dust number concentration on the order of $0.1\,\mathrm{cm}^{-3}$, i.e. comparable to the results obtained for the ICE-L case. The modification of the water and aerosol profiles depends also on the chosen parameterisation of homogeneous freezing and the parameterisation of hydrometeor fallspeeds. The impact of altering these parameterisations have not been tested in the present study, but should be investigated in future work. Based on the idealised KiD simulations we develop a conceptual model that depends on the potential condensate, in-cloud residence timescale, deposition timescale and sedimentation timescale. Lagrangian estimates of the latter two timescales are used to derive an approximation to the timescales, while the other necessary variables can be calculated analytically from the upstream thermodynamic and aerosol profiles. The resulting model captures the variability of the downward transport of moisture by sedimenting hydrometeors in a large part of the phase space with deviations less than $30\,\%$ for almost all parameter combinations. The error is somewhat larger for cloud top temperatures between $-36\,^{\circ}\mathrm{C}$ and $-42\,^{\circ}\mathrm{C}$, i.e. in the transition region between clouds dominated by heterogeneous and those dominated by homogeneous freezing.

The analysis in the present paper suggests that UM-CASIM framework can reasonably capture some key components of mixed-phase orographic clouds such as the vertical velocity structure, the co-existence of liquid and ice particles, and the existence of regions dominated by ice crystals formed by heterogeneous or homogeneous freezing. However, there are some deviations between the modelled and observed thermodynamic conditions and cloud properties. These deviations maybe do to spatiotemporal variations in the upstream thermodynamic fields and the structure of the wave, which are not well characterised in the available observational data. The deviations may also result from uncertainty in the regional model predictions due initial and boundary condition uncertainty. And finally errors in the model representation of dynamics and subgrid-scale processes may be the source for the differences between observations and model results. It is important to properly explore all these options, which is beyond the scope of the present paper but will be addressed in future work. As the UM-CASIM simulations can currently not be vigorously constrained with observations, there is also some uncertainty as to the accuracy of the idealised, two-dimensional simulations and the derived conceptual model. This pertains mainly to the formulation and absolute values of the timescales.

While it is not possible to constrain the downward transport of aerosol or water vapour with the observations available from ICE-L, future aircraft campaigns targeting orographic wave clouds would be useful to quantify these important processes and provide constraints on aerosol transport processes also for more comprehensive aerosol models such as UK Chemistry and

Aerosol Model (UKCA, e.g. Planche et al., 2017). Any future campaign should aim at a better characterisation of the the upstream and downstream moisture and aerosol profiles including their temporal evolution and a characterisation of the 3D velocity field. The idealised simulations show that clouds with wave periods larger than $1000\,\mathrm{s}$ and cloud top temperatures between $\sim -19\,^{\circ}\mathrm{C}$ and $\sim -28\,^{\circ}\mathrm{C}$ ($\sim -30\,^{\circ}\mathrm{C}$ and $\sim -38\,^{\circ}\mathrm{C}$) show a large sensitivity of the downward aerosol (humidity) transport to choices in the cloud microphysical parametrisation. Similarly, differences between simulations with different heterogeneous freezing parameterisations are largest for wave periods larger than $800\,\mathrm{s}$ and cloud top temperatures between $\sim -30\,^{\circ}\mathrm{C}$ and $\sim -38\,^{\circ}\mathrm{C}$. These regions of the phase space therefore would be interesting to target in future observational campaigns.

*Code availability.* The source code of the KiD-A model version used here and the namelist files are archived in a private directory on `bitbucket.org` (`amiltenberger/kida_model_with_taubin_casim_icel`). Access will be granted by the authors on request.

*Data availability.* Model data are stored on the tape archive provided by JASMIN (http://www.jasmin. ac.uk/) service and can be provided by the author on request.

*Author contributions.* All authors contributed to the development of the concepts and ideas presented in this paper. A. A. Hill developed the KiD model. A. K. Miltenberger developed the model set-up with major contributions from P. R. Field for the UM simulations and A. A. Hill for the KiD set-up. P. R. Field and A. J. Heymsfield provided expertise on the observational data. All model simulations and the subsequent data analysis were performed by A. K. Miltenberger. She also wrote the majority of the manuscript with input and comments from all co-authors.

*Competing interests.* The authors declare that they have no conflict of interest.

*Acknowledgements.* We thank Ben Shipway and Jonathan Wilkinson for designing CASIM and incorporating it into the UM model. A. Miltenberger also thanks Axel Seifert for insightful discussions on the characteristic timescales of ice-phase orographic clouds. We acknowledge the efforts of the ICE-L team for obtaining the observational data. Further, we acknowledge use of the Monsoon2 system, a collaborative facility supplied under the Joint Weather and Climate Research Programme, a strategic partnership between the Met Office and the Natural Environment Research Council. All KiD simulations and data analysis were performed on JASMIN, the UK's collaborative data analysis environment (http://jasmin.ac.uk).

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

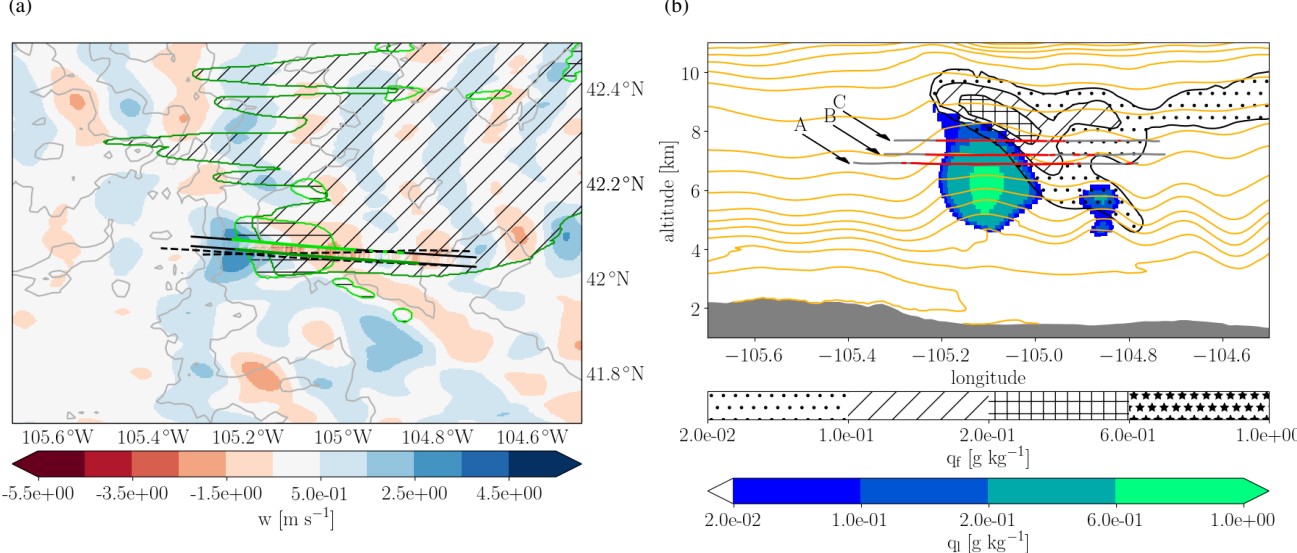

**Figure 1.** (a) The modelled vertical velocity field at $7200\,\text{m}$, i.e. approximately the altitude of flight leg B, is shown by the colour shading (2100 UTC). The dark blue (cyan) contour and horizontal (diagonal) hatching indicates where liquid (frozen) cloud water content in the model exceeds $0.02\,\text{g}\,\text{kg}^{-1}$. The color shading in the area between the two grey straight lines shows the observed vertical velocity along flight leg B and the blue (green) colouring of the grey lines indicate observed cloud liquid (ice) exceeding $0.02\,\text{g}\,\text{kg}^{-1}$. The black dashed lines show the location of the aircraft legs at $6900\,\text{m}$ and $6780\,\text{m}$. Black contours indicate the topography. (b) Vertical cross-section through the wave cloud at $42.05\,^\circ\text{N}$ (2100 UTC). The colour shading represents the modelled liquid water content, the contour lines with the hatching the modelled ice water content and the orange lines indicate isentropes. The horizontal lines show the projection of the flight path on the plane of the cross-section, where red colouring of the lines indicates observed cloudy conditions (condensed water content larger than $10^{-7}\,\text{kg}\,\text{kg}^{-1}$). The grey area at the bottom of the plot shows the topography.

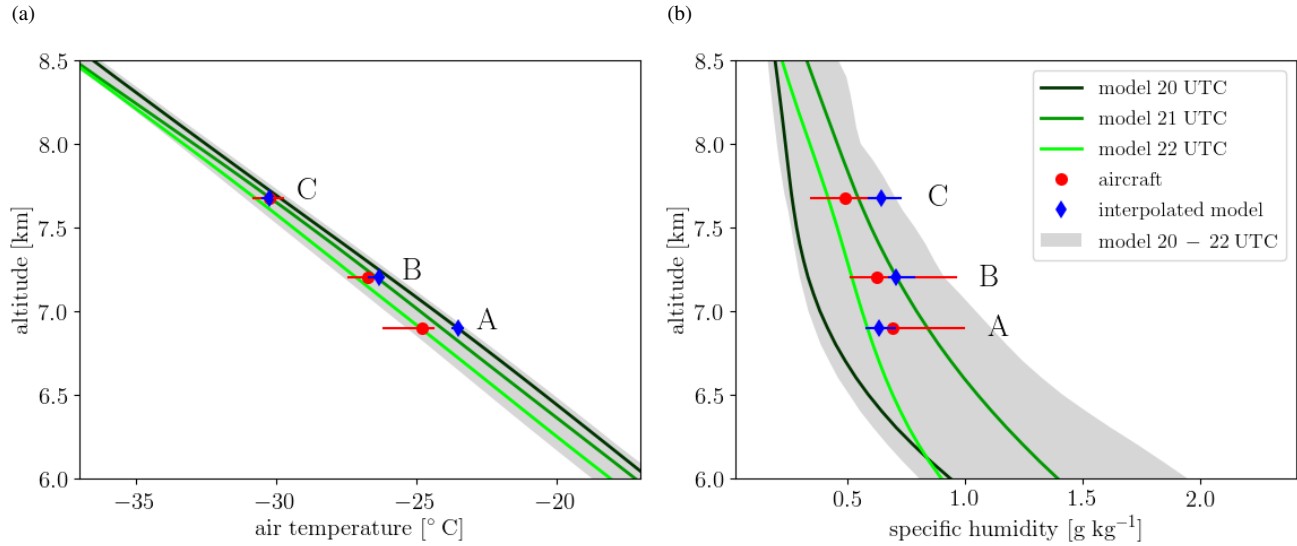

**Figure 2.** Comparison of upstream temperature (a) and specific humidity (b) profiles from the UM simulation and aircraft data. Upstream conditions from aircraft data are computed from the non-cloudy sections of the aircraft legs west of $-105\,^\circ$E. Red circles indicate the mean value along these portions of the aircraft legs and the bars the variability. The model values are taken from the grid column closest to the average location of these upstream aircraft segments (green lines) at times between 2000 UTC and 2100 UTC, i.e. bracketing the time of the observations between $\sim 2040$ UTC and $\sim 2120$ UTC. The cyan shading shows the variability of temperature and specific humidity in this grid column for all output times between 2000 UTC and 2200 UTC. The blue diamonds and bars show the model data interpolated to the flight track and evaluated in the same way as the aircraft data.

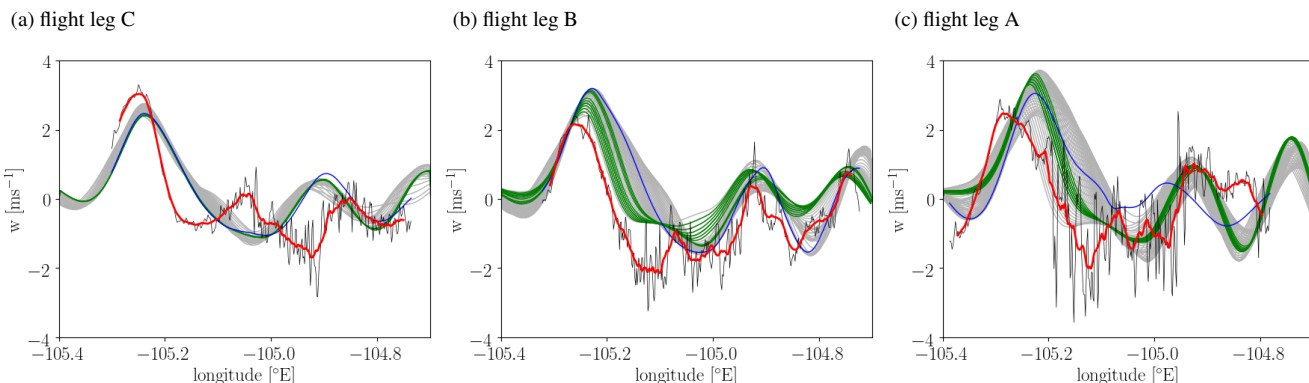

**Figure 3.** Vertical velocity along flight leg C (7680 m, 2120 UTC) (a), flight leg B (7200 m, 2100 UTC) (b) and flight leg A (6900 m, 2040 UTC) (c). The red solid line shows the aircraft data smoothed with a 20 s moving average filter (full 1 Hz data shown by the thin black line). The blue line shows the modelled aircraft velocity interpolated to the aircraft track. The grey lines show the vertical velocity along tangents to the mean streamline (including a deviation corresponding to the deviation between the observed mean horizontal wind direction and the direction of the aircraft track), for which the peak vertical velocity exceeds $2.5\,\mathrm{m\,s^{-1}}$. This threshold was chosen to focus on the centre of the wave cloud only. The green line shows the tangent for which the Pearson correlation (including a lag of $\pm 20\,\mathrm{s}$) with the observed vertical velocity is larger than 0.95. To account for the temperature bias of the model, model data are taken 200 m above the altitude of the flight track.

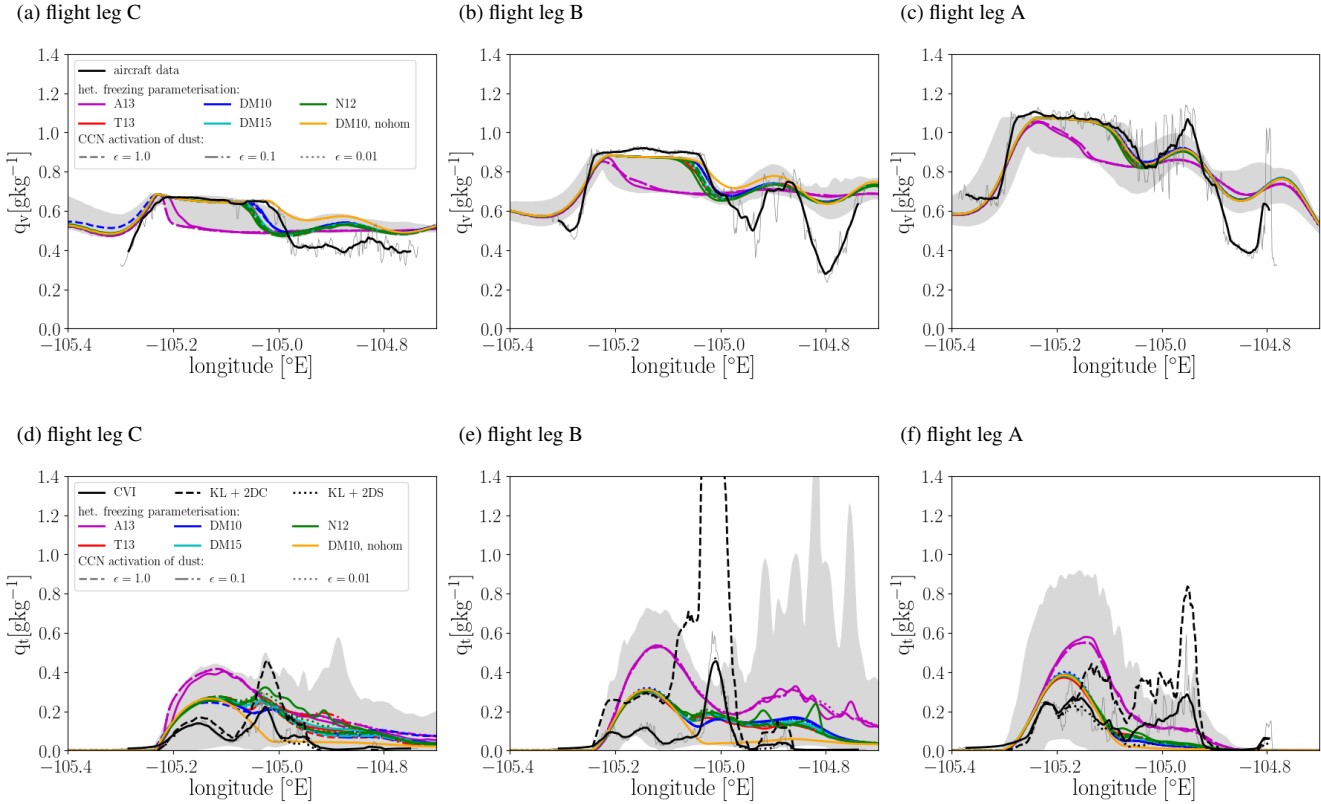

**Figure 4.** Comparison of specific humidity (a-c) and total condensation mass mixing ratio (d-f) for the three different flight legs. The flight legs are shown in the sequence of decreasing flight altitude from left to right. The thick black lines shows the smoothed aircraft data (thin black line shows 1 Hz data). For the total water content (d-f) data from the CVI (thick black solid line) as well as the sum of King liquid water probe data and 2DS (2DC) data (black dashed (dotted) line) is shown. Model data are interpolated to the same tangents of the mean streamlines as used in Fig. 3. With the exception of the CVI data, the observed total water content includes only ice crystals larger than 50 $\mu$m. From the model results, the ice water content for ice crystals larger than 50 $\mu$m is computed by integrating over the respective part of the assumed size distribution in the model using the prognostic variables of ice number concentration, ice mass mixing ratio and the fixed shape parameter. The modelled variability of the variable along all these hypothetical flight paths are shown by the grey shading, while the thick coloured lines show the median values for simulations. The different coloured lines represent with different ice nucleation schemes and different line styles indicate different assumptions on the amount of soluble material in the dust particles (solid: all dust acting as INP).

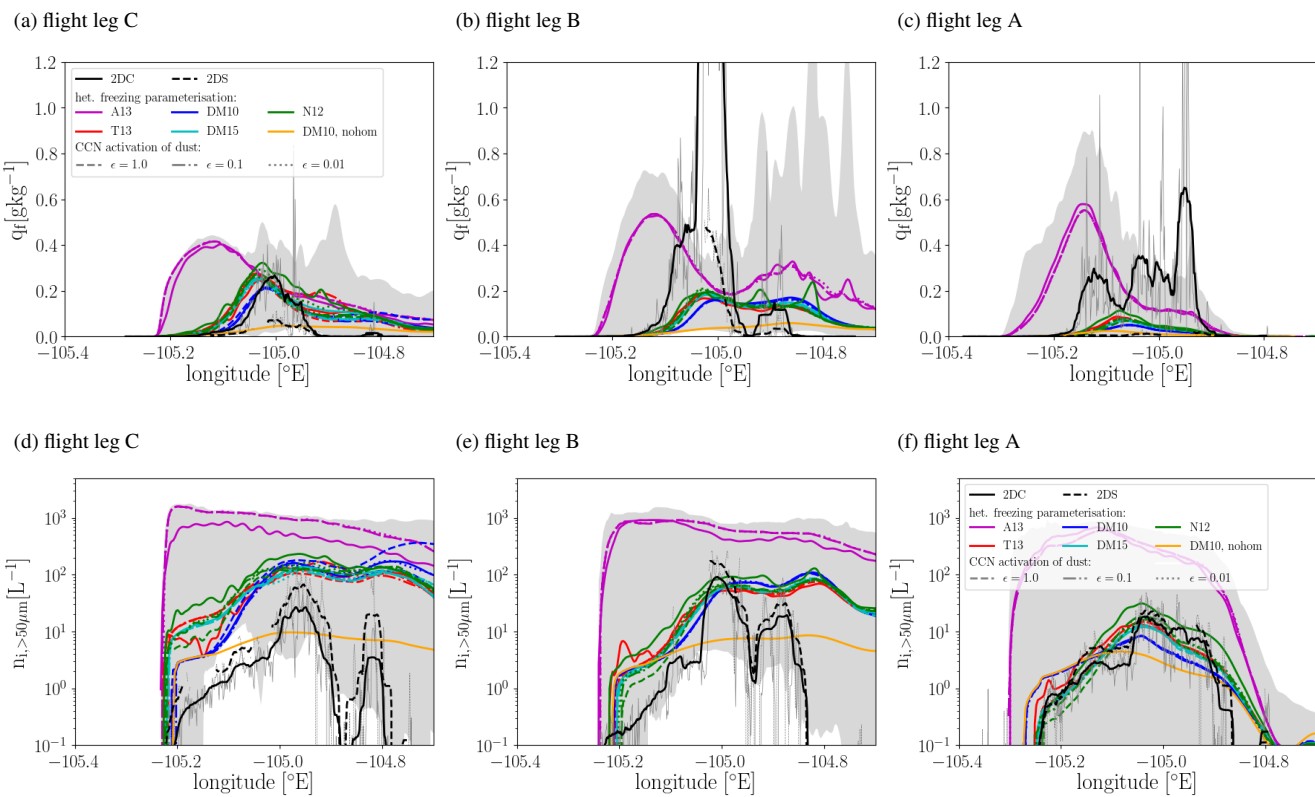

**Figure 5.** As Fig. 4 but showing the frozen hydrometeor mass mixing ratio (a-c) and ice crystal (d-f) number concentration. The data (mass mixing ratios as well as number concentrations) incorporate only crystals larger than $50\,\mu$m.

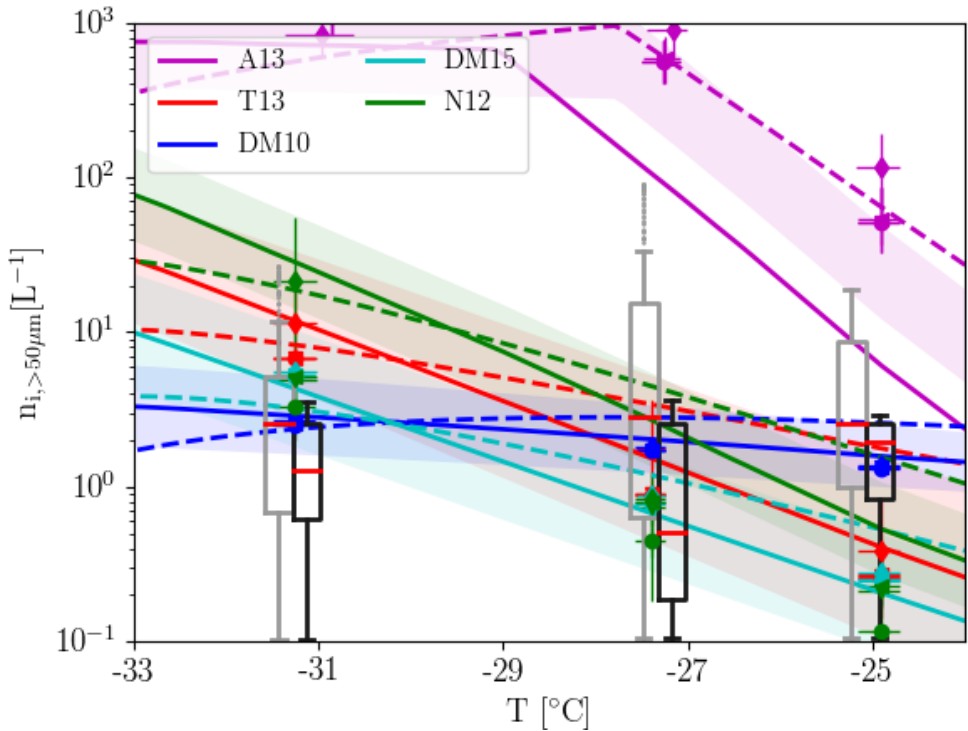

**Figure 6.** Temperature dependency of ice crystal number concentration against air temperature from observations (2DC, box-plots) and UM model simulations (symbols, colours represent simulations with different ice nucleation schemes according to legend in Fig. 4 a). Model data are interpolated to the hypothetical flight tracks and only considered in the first part of each flight leg, i.e. the updraft region. Observational data are also sub-sampled to include only data from the updraft region, which are shown in the black box-plots (grey box-plots show all data). The solid lines show the expected ice crystal number concentration based on ice nucleation only using the prescribed dust profiles. The colour shading illustrates the expected ice crystal number concentration for dust number concentrations within a factor 2 of the used profile, i.e. compatible with range observed upstream of the cloud. Assuming a linear decrease of the upstream dust concentration over the time period of the observations together with the assenting flight pattern results in expected ice crystal number concentration as shown by the dashed lines.

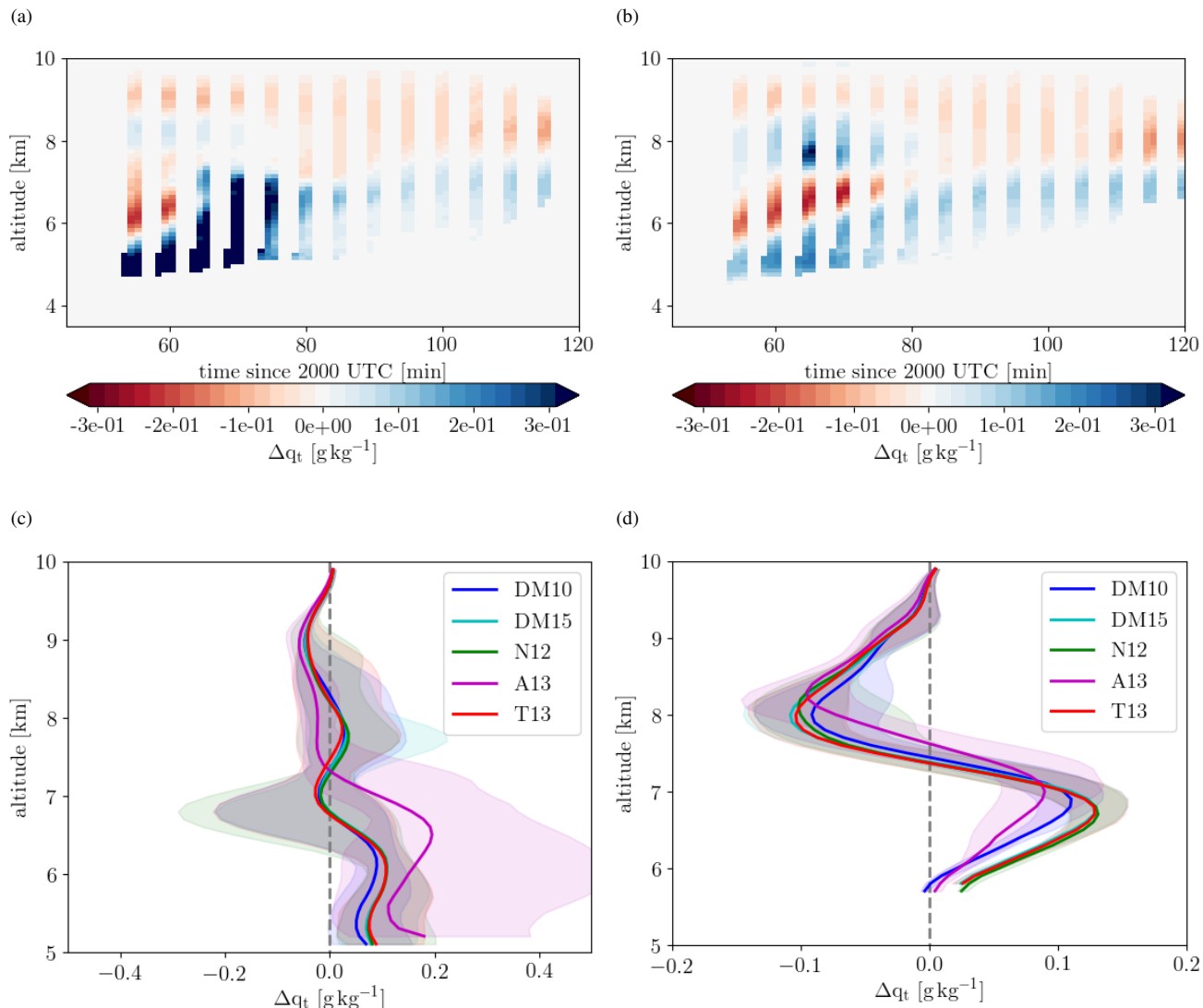

**Figure 7.** (a,b) Difference in total water mass mixing ratio $\Delta q_t$ between $-105.35\,°\text{E}$ (upstream) and $-104.78°\,\text{E}$ (downstream) along backward trajectories for simulations using the A13 (a) and the (b) DM10 ice nucleation parameterisation, respectively. The difference are calculated as downstream values minus upstream values. The plot shows values at $42.1°\,\text{N}$, i.e. downstream of the centre of the wave cloud. The time on the abscissa indicate the arrival time of the trajectories at the downstream location. (c,d) Mean profiles of $\Delta q_t$ for all simulations averaged between $2110 - 2130\,\text{UTC}$ (c) and $2140 - 2200\,\text{UTC}$. The different colours correspond to simulations with different ice nucleation parameterisations, while the shading represents the temporal variability of the profiles. Note the travel time of the trajectories between the upstream and the downstream location is about $30 - 40\,\text{min}$.

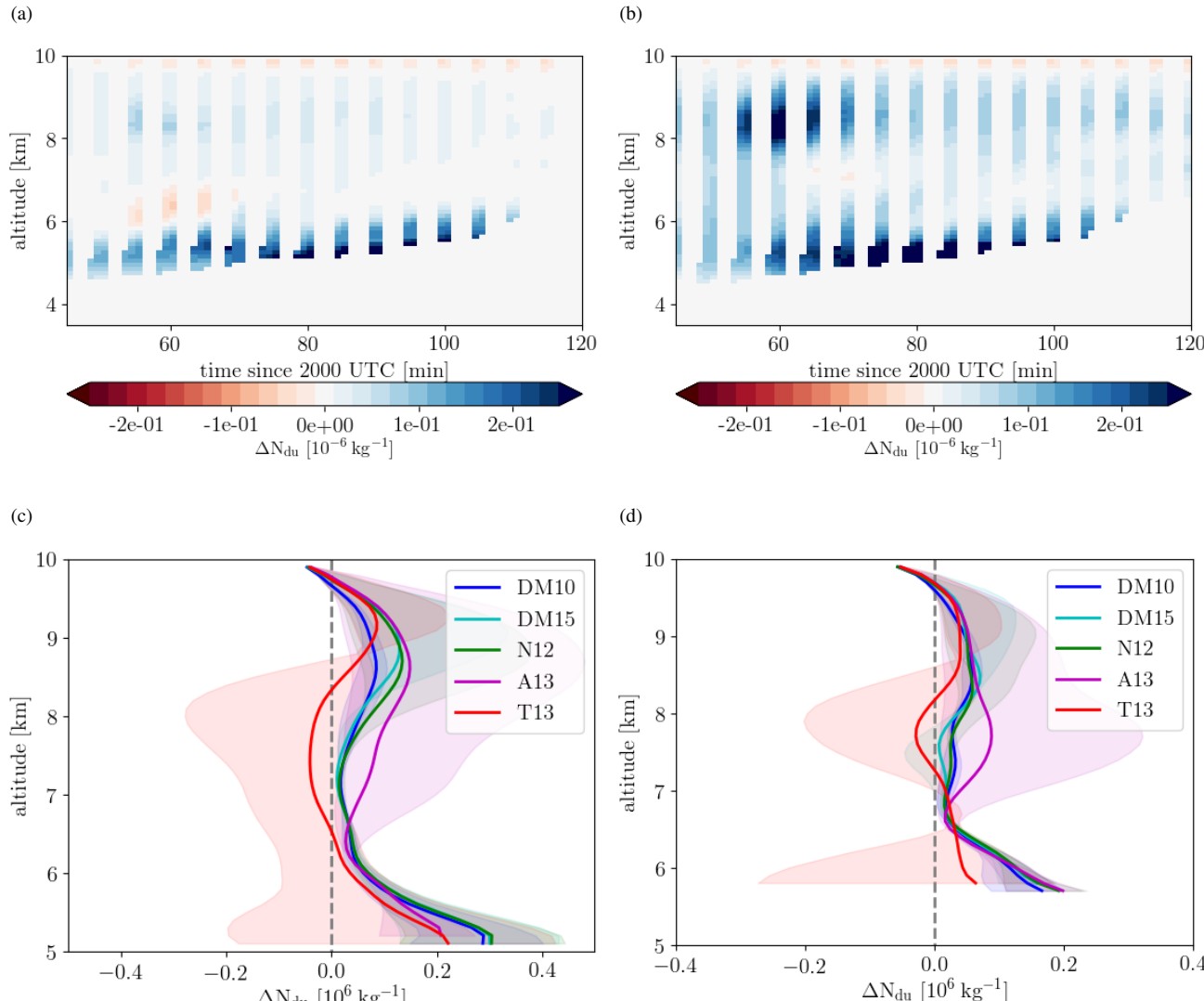

**Figure 8.** As Fig. 7 but showing the change in dust number concentration.

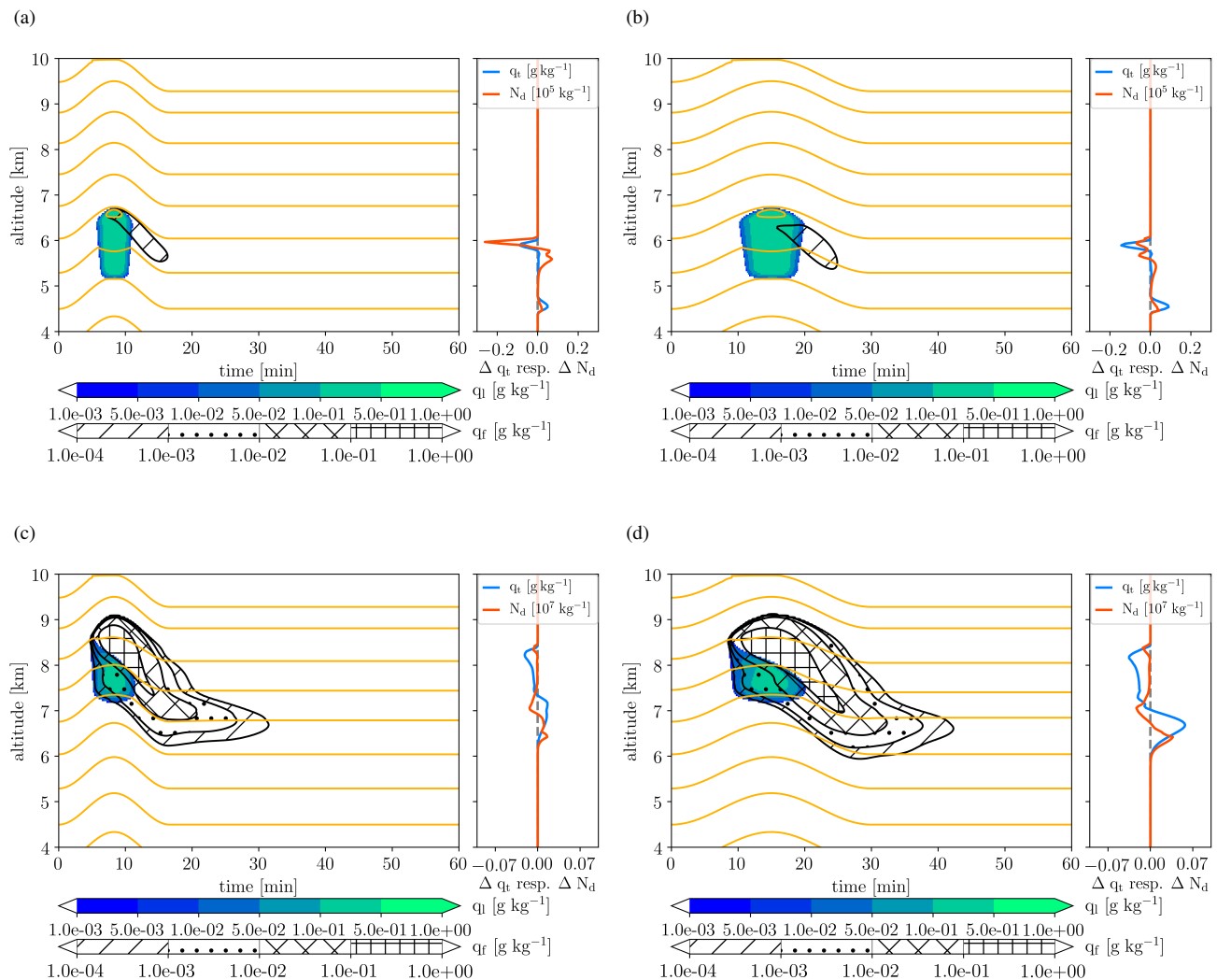

**Figure 9.** Wave clouds in the KiD-model. Simulations of waves with periods $1000\,\mathrm{s}$ (a, c) and $1800\,\mathrm{s}$ (b, d) are shown. The cloud top temperature is $-24\,^\circ\mathrm{C}$ in panels (a, b) and $-50\,^\circ\mathrm{C}$ in panels (c, d). The cloud droplet mass mixing ratio is indicated by the colour shading, ice and snow mass mixing ratio by the hatched contours and the isentropes by orange isolines with a spacing of $2\,\mathrm{K}$. The small sub-panels show the difference in total water content (light blue line) and the dust number concentration (red line) between the upstream and downstream. Note the different units for the dust number concentration change in panels (a, b) and (c, d), respectively.

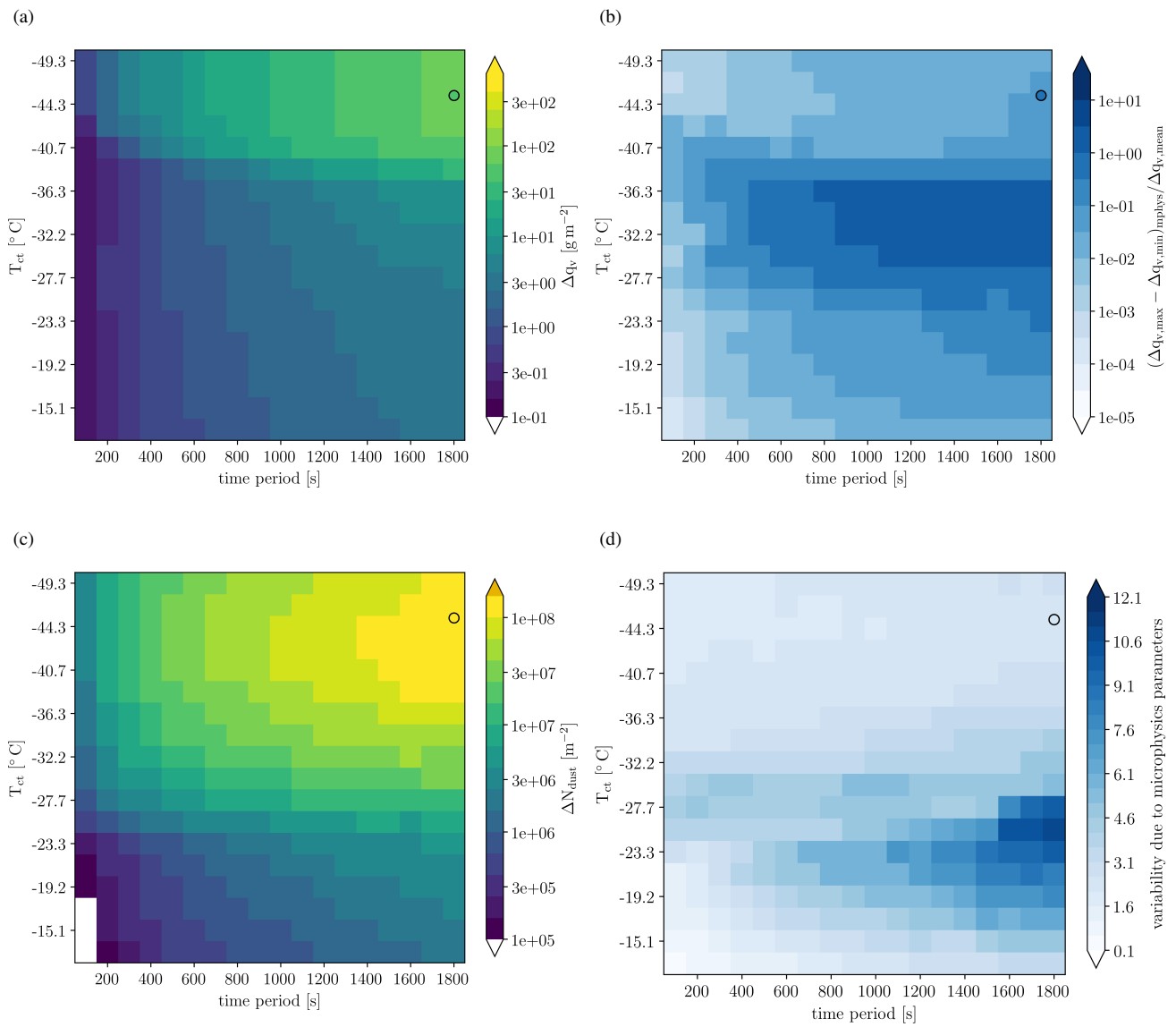

**Figure 10.** Modification of water (a, b) and dust (c, d) profiles across wave clouds with vertical extend $z_c$ of 2 km and various cloud top temperatures (ordinate) as well as periods (abscissa). The panels (a) and (c) show the mean value across KiD-simulations with different ice nucleation and soluble fraction descriptions. The panels (b) and (d) show the variability resulting from varying the ice nucleation representation and the soluble fraction assumption, i.e. $(\Delta_{\mathrm{Lagr}} q_t|_{\max} - \Delta_{\mathrm{Lagr}} q_t|_{\max})/\Delta_{\mathrm{Lagr}} q_t|_{\mathrm{mean}}$ . The colour-filled circles indicate the location of the ICE-L case study in the phase-space. The color of the circle showsn the value obtained from the UM simulations of the ICE-L cloud.

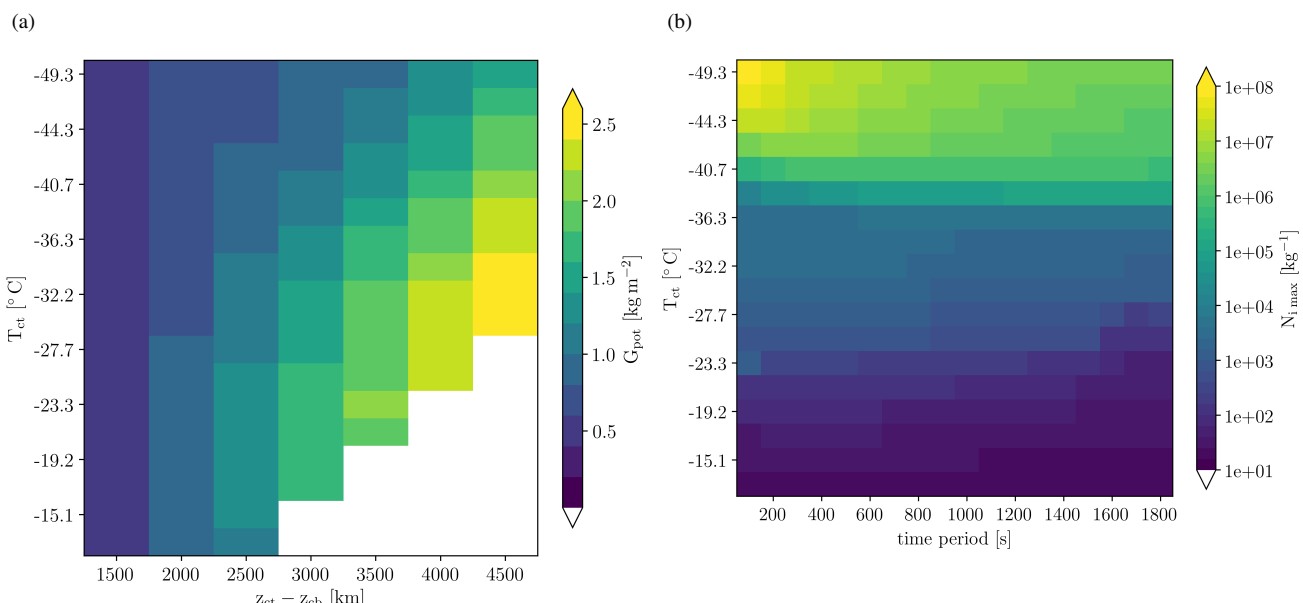

**Figure 11.** (a) Potential condensate $G_{pot}$ as a function of cloud thickness and cloud top temperature. (b) Maximum ice crystal number concentration $n_{i,max,Lagr}$ as a function of wave period and cloud top temperature for clouds with a thickness of $2\,km$. $n_{i,max,Lagr}$ is the maximum integrated ice crystal formation rate, including homogeneous and heterogeneous freezing, along any trajectory through the wave cloud.

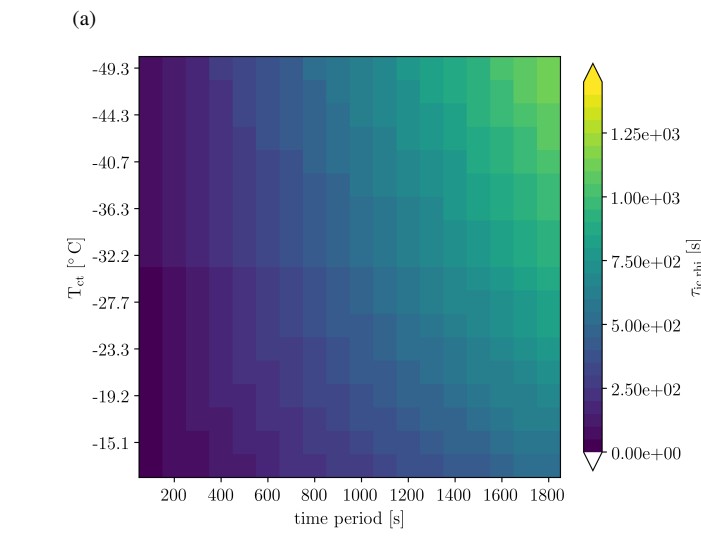

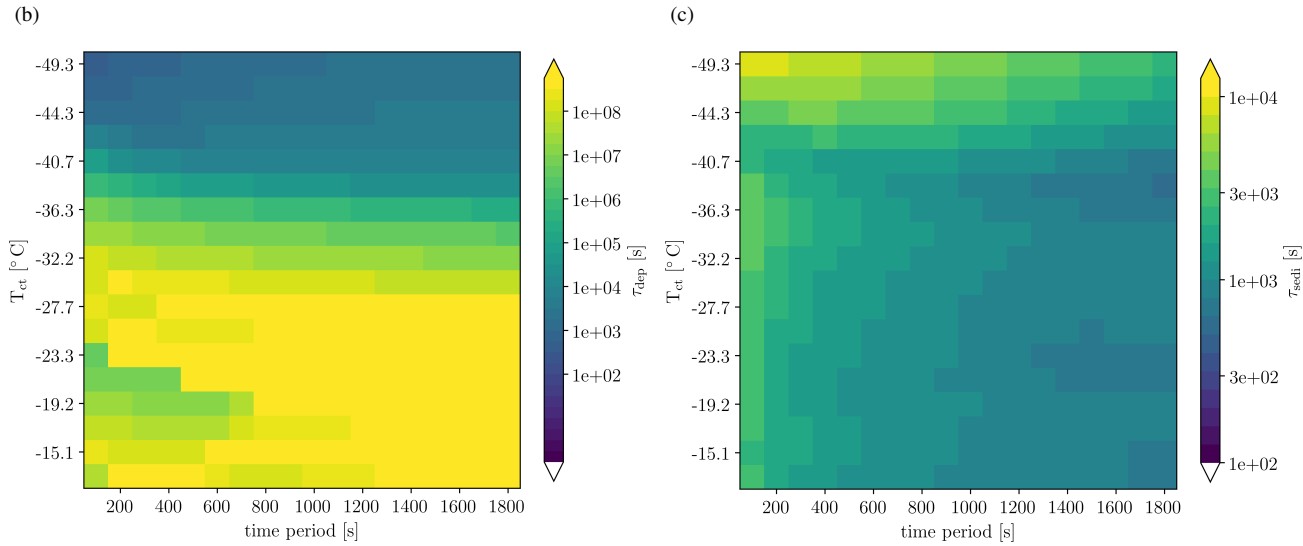

**Figure 12.** Lagrangian estimates of (a) the in-cloud residence timescale $\tau_{ic}$, (b) the deposition timescale $\tau_{dep}$ and (c) the sedimentation timescale $\tau_{sedi}$. Results are shown for simulations with a cloud thickness of 2 km.

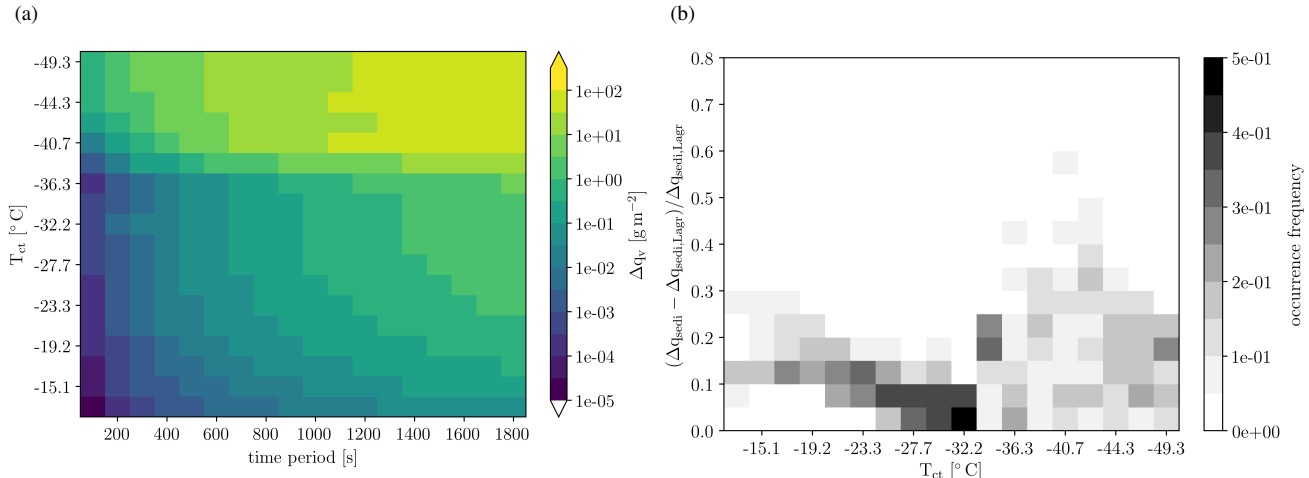

**Figure 13.** (a) Total downward transport of water predicted using equations 3 to 8 across wave clouds with vertical extend $z_c$ of 2 m, the DM10 heterogeneous freezing parametrisation, and various cloud top temperatures (ordinate) as well as periods (abscissa). (b) Normalised difference between $\Delta q_t$ predicted by the conceptual model and the full KiD model for different cloud top temperatures. The data shown in (b) includes the full simulations set with all cloud top temperature, wavelength, and cloud thickness specified in section 2.3.