# Peer review of "Vertical redistribution of moisture and aerosol in orographic mixed-phase clouds"

_Atmospheric Chemistry and Physics, 2019_

## Referee Comment (RC1) · Anonymous Referee #2 · 8 Feb 2020

This study presents an evaluation of the Unified Model (with the CASIM module) using wave cloud aircraft observations. Multiple heterogeneous freezing parametrizations are examined and compared with the observations. In addition, a large number of the KiD model simulations are used to examine the impact of the mountain wave period and cloud-top temperature on the cloud evolution. Finally, the authors provide a conceptual model based on the acquired knowledge, which estimates the wave cloud-driven redistribution of water vapor.

I find this manuscript quite comprehensive and well written. I like the conceptual model idea, although it is obviously case-dependent (as suggested by the authors as well). However, I think that additional work needs to be done before this manuscript can be published in ACP (Major revisions).

[Figure]

Major comments: - UM results: I am not convinced that there is a very good agreement between the UM and the observations (l. 239-244). The statement in the text is subjective, that is, in terms of percentage, the specific humidity errors are very large, on the order of up to several tens of percent (see fig. 2b). Same for the vertical velocity amplitude - errors of 1 m/s can be on the order of ~50% relative to the observations (e.g., leg 2 – fig. 2b). As the UM simulations serve as a benchmark for the KiD simulations, the implications of a weak agreement between the UM and the observations could be significant.

- Homogeneous freezing regime: given the fact that for a large part, the heterogeneous freezing parametrization is examined here, the UM simulations, and hence, also a large fraction of KiD simulations, are "contaminated" by homogeneous freezing in the top of the cloud layer (e.g., fig. 7), which obviously non-linearly impacts the underlying heterogamous cloud layer. The authors did not refer to the homogeneous freezing parametrization in the UM and how well it corresponds with the parametrization in the KiD model. Now, I presume that the UM is too complicated to vary the homogeneous freezing parametrization, but I suspect that the influence of other parametrizations can be examined in the KiD model. An additional approach to address large parts of this comment would be to run the UM initialized without homogeneous freezing influence on the cloud layer (e.g., by offsetting the temperature profile to higher values while retaining the RH profile). I wonder how much would the results of this study change in that case (e.g., deviations of the UM from the observations)?

- Deposition nucleation is not mentioned at all in the text, although the current understanding is that its efficiency significantly increases when we approach the homogeneous freezing regime (e.g., Kanji et al., 2017, https://journals.ametsoc.org/doi/full/10.1175/AMSMONOGRAPHS-D-16-0006.1).
I understand that we would typically except immersion freezing to still account for most of the nucleation, but the deposition mode should still be mentioned in the introduction as well as in the model description, even if it is eventually omitted from the simulations.

- Negligible impact of rain processes in the KiD simulations and neglecting rain processes in the conceptual model: the minor impact of rain could be driven by: a. the dominance of the homogeneous ice precipitation from cloud top, b. high cloud droplet number concentration that leads to weak collision-coalescence in the model, and/or c. influence of the collision-coalescence kernel implemented in the model. I am not convinced that rain processes are indeed negligible and whether they should be neglected in the conceptual model. If necessary, I presume that adding a "rain component" to the conceptual model should not be a difficult task, given previous conceptual models for warm clouds (l. 435-436).

- Importance of periods longer than 1000 s (stated in the abstract and l. 554-557): do not believe this conclusion, as the authors did not consider different obstacle heights (eta values I presume). As a result, the impact of the (likely maximum) vertical velocity is nearly ignored in the parameter space evaluation, although it should definitely impact the longevity of the wave cloud, among other factors, via ice processes, especially in the homogeneous freezing regime, where we would expect to lose all condensated droplets relatively fast.

Minor comments: l. 49-52 – That is a rather complex sentence. I suggest rephrasing or breaking into two separate sentences.

l. 109 – ICE-L acronym definition is missing.

l. 109 - please add parentheses to the citations.

l. 109 – inconsistent date (November 17th) with the following sections and figures.

l. 127-130 – These two sentences seem redundant - repeating information already provided in the Introduction.

l. 146 - Suggest consistency in the number of fractional coordinate digits.

l. 147 – focusses –> focuses

l. 167 - Please define the source for this eta value - I will presume that it is equivalent to the obstacle (mountain) height

l. 168 – 32.1 K - Suggest consistency about the temperature units (C instead of K).

l. 173 – Eq. 2 I do not find consistency in the definition of the different z ranges.

l. 175 – (nothing to revise here) – Figure 9 is very nice.

l. 201-202 – What is the parametrization for ice hydrometeor fall velocity used in UM and KiD? This may have a substantial impact on the results presented below.

l. 242 – I presume "basis" should be "bias"

l. 252-253 - If the correction is made for 0.02 g/kg, what is the importance of 0.0001 g/kg in fig. 1?

l. 282 - I can only see some sort of an ni agreement in fig. 5f. I can't interpret the max ni intersection in panel e as model-observations agreement. Using that parameter as a measure of model performance could be misleading.

l. 321 - "data .. is . . ." - "Data" is the plural form of "datum" - please correct the text and figure captions accordingly (is –> are, etc.)

l. 330 - best agreement with DM10 and TB13 - how do you define the best agreement? In some aspects (e.g., absolute ni values), the log scale in fig. 6 is misleading because I would suggest that the best agreement is with DM10 and DM15.

l. 365 - 0.1 g/kg - these are mixing ratio units, not flux units. I suggest providing a more consistent terminology throughout the text (also related to the conceptual model).

l. 403 - a minus sign is missing for the temperature

l. 406 - I suggest adding a reminder to the reader about which two simulations are discussed here.

l. 445-446 - I presume that the definition of Gpot is the integration of qv minus qs(Tmin),

yes? Please clarify, or alternatively, describe eq. 4 earlier in the text.

l. 473 – "saturation)" - redundant parenthesis

l. 479 – inline equation - not all terms in this equation are defined.

l. 492 – pre-scribed –> prescribed

l. 500-501 & Fig. 13 - I suggest adding a panel that shows the difference between the conceptual model and the KiD model. The discrepancies I see in Figure 5 in the SI (especially in panels c and d, which are shown in log scale) suggest that the differences can be quite significant when the time period and Tct parameter space are examined. Also, shouldn't there be consistency regarding the discussion/use of total water (qt) and water vapor (qv) throughout the text in figures (7-13)?

l. 504 - I presume that 'b' is missing when referring to fig. 13.

l. 538 - Except for the conceptual model, I did not encounter any discussion about the results based on varying cloud thickness. I suggest adding some information to the text and figures, or removing it from this discussion about the conclusions.

Fig. 1 - Suggest changing the cyan, blue, and gray curve colors - they mixed with the colormap. Also, the units in the colorbar are confusing here. Also, What altitude do the black contours represent? This should be specified.

Fig. 1 and discussion in the text - Is $10^{-7}$ kg/kg above the aircraft instrumentation uncertainty level for IWC and LWC? This should be discussed and justified in the text, i.e., what is the "true" extent of this cloud field given measurable justified quantities?

Fig. 2 - I suggest redefining the altitudes for each flight leg

Fig. 4 - I'm having a tough time reading the axes labels

Fig. 3-5 - please provide a title for each panel stating the flight leg and/or altitude. At the moment, it is hard to follow the text.

Fig. 5-6 – suggest adding a legend instead of directing the reader to fig. 4 every time.

Fig. 7 - blue for positive values in panels a,b is counter-intuitive. I suggest flipping this colormap.

Fig. 9 - Please correct qv–>qt in the legend

Fig. 10 - The figure caption is not complete, e.g., last sentence, circle markers, the definition of the two simulations, etc. The varying contour colors in panels b,d are quite confusing.

---

## Referee Comment (RC2) · Minghui Diao (Referee) · 23 Feb 2020

Reviewed by Minghui Diao

This manuscript uses the ICE-L field campaign measurement to compare with simulations from the Unified Model. In addition, idealized simulations from a 2-D model, the KiD model, were used to conduct additional simulations and examine downward moisture flux. The UM model included a recently developed module – CASIM, which enables the analyses of dust particles and their impacts on liquid and ice hydrometeors. The overall organization of the writing is straightforward. The sensitivity tests on various heterogeneous nucleation parameterisations provide valuable information. The reviewer has a few major comments, followed by some minor comments. A major

revision is recommended before being considered for publication at ACP.

1. For the ICE-L field campaign, the 2DC data are restricted to > 125 micron. However, it is not clear if the model outputs of ice water content (IWC) have considered the size cutoff in ice crystal size distribution. From Figure 5 axis label and caption, it seems that ice crystal number concentration (Ni) has been restricted to > 125 micron. But in the legend of Figure 4, qt did not mention any size cutoff. Please clarify this in the main text besides the figure legend.

Also, can the authors comment on possible impacts of the comparison results if small particles were included in the comparison? For example, would the model show better or worse results compared with observations?

2. Another main comment is related to measurements of 2DC and CDP. In Line 133 – 135, the authors commented that 2DC probe is used for IWC and Ni, and CDP is used for cloud number concentration (I assume that you mean liquid droplet number concentration?). However, in previous studies, we found that 2DC may measure some large drizzles, while CDP may measure some small ice. A detailed discussion about separating liquid and ice from 2DC and CDP measurements was given in D'Alessandro et al. (2019, J. Climate), https://doi.org/10.1175/JCLI-D-18-0232.1, "Cloud Phase and Relative Humidity Distributions over the Southern Ocean in Austral Summer Based on In Situ Observations and CAM5 Simulations".

Can the authors comment on the potential impacts on the model evaluation, if some ice was misidentified as liquid in CDP measurements, and some liquid was misidentified ice in 2DC measurements? Some sensitivity tests on possible variations of IWC, LWC, Nice and Nliq derived from field observations would be helpful.

3. Homogeneous freezing has been briefly mentioned in a few places, but there are not many discussions on the quantitative impacts from it compared with heterogeneous nucleation. For example, even though homogeneous freezing is more dominant at colder temperatures, such as at below -37 C, ice crystals formed by homogeneous freezing can sediment into lower altitudes, and therefore being misidentified as ice formed via heterogeneous nucleation. Can the authors comment more on this sedimentation effect?

The reviewer suggests that the authors quantify IWC and Ni into two categories – those originated from homogeneous freezing versus those from heterogeneous freezing. Would this be possible for the model used here? For example, additional lines can be added to Figure 4 (d-f) analysis of qt and Figure 5 (d-f) analysis of Ni, to quantify these two components.

4. In the conclusion and the result section, when comments were made on whether the model performance is good or not, it seems a little arbitrary. One suggestion is to add some comparisons with previous studies, or with the older versions of the same model. If improvements are seen compared with previous work, then it is more convincing that this model performs better.

Below are some minor comments:

Line 199, "the he modification". Typo.

Line 161, recommend adding a full description of notations for temperatures and altitudes used in this study. For example, there is $t\_ct$ for cloud top temperature, but later in Figure 10, the axis label uses $T\_min$ for cloud top temperature. Please be consistent. Cloud thickness is defined as $z\_c$, but the definitions of $z\_ct$ and $z\_cb$ is not explicitly mentioned (I assume they are cloud top and cloud base height, respectively).

In equation (2), there are notations of $z\_ct,t$ and $z\_cb,t$. How are they different from $z\_ct$ and $z\_cb$?

Line 168, "32.1 K" should be in Celsius.

Line 243, "temperature basis", biases?

Line 275, the authors mentioned that "while significant cloud glaciation also only oc-
curs in the downdraft region, ice crystal number concentration increases further downstream". Is there any explanation why significant cloud glaciation only occurs in the downdraft region? It seems counter intuitive that downdraft leads to glaciation and new ice crystal formation.

Line 277, "in the model the air parcels likely experience larger vertical displacement", is there any evidence of the parcel displacement? Is it possible that other factors could lead to higher ni, such as homogeneous freezing is being activated too early, allowing too little clear-sky ice supersaturation?

Line 278, "ice crystal population at observed along flight legs", delete "at observed"?

In the same line, "a earlier", an earlier.

Line 279, "... ice crystal number masking the depositional growth", should it be "ice crystal number <and> masking the depositional growth"?

Line 292 - 293, "the longevity of ice crystal... related to smaller average ice crystal mass...", what is the meaning of ice crystal mass? Do you mean the mass of individual ice crystals, or the total ice water content?

Line 298 – 299, "the overestimation in initial ice crystal number is either related to the heterogeneous freezing parameterisations used or a too large diameter of the newly formed ice crystals." Heterogeneous freezing generally forms fewer ice crystals than homogeneous freezing. Is it possible that the high ni here is contributed by homogeneous freezing? In addition, the comment on the model having too large ice crystals and therefore overestimating Ni doesn't seem right. If the diameters of the newly formed ice crystals are too large, they would sediment faster and reduce the ice crystal number concentration. In addition, if the total water content is conserved, forming too large ice crystals would lead to fewer ice crystals, not more ice crystals.

Line 305, "observations if", observations of?

Line 311, "This data". Data should be in plural form. This typo occurs in several places,

including figure captions and the "data availability" section. Please use a global search to correct them all. Same for Line 321, observation data . . . is, should be are.

Line 328, "but introduces", and introduces?

Line 330, here both DeMott et al. (2010) and Tobo et al. (2013) are mentioned as the ones giving the best agreement. But in the conclusion section, only DeMott (2010) is mentioned. Maybe the conclusion can provide more comments on the best agreement based on specifically what variables.

Line 398, -45 deg c, "c" should be C.

Line 403, 37 deg C should have a minus sign.

Line 451, the equation (t + A*gamma) should be (t0 + A*gamma)? If not, what is "t" here?

Line 459, please add a comma between "clouds" and "reflecting". Some other sentences are too long as well without a comma to separate different parts of the sentences.

Line 464, several log_10 didn't have the 0 in subscript.

Line 476, w = 0 ms, should be m s-1.

Line 479, please clarify the meaning of each term in the equation.

Equation 6. K should be deg C

Equation 7. K should be deg C. Also, there is a km unit. Should be C?

Line 511 – 512, this would be a good place to add comments on previous model evaluation studies and compare with the results shown here.

Line 512, 1 ms, should be 1 m s-1.

Line 539, -30 K and -40 K, should be deg C.

Line 541, 0.1 g kg, should be g kg-1.

Figures 1, 2, 3. Suggest adding labels to three segments as A, B, C, and use texts and arrows to highlight them in Figure 1b. It would make it a lot easier to match them with the figure legends and lines in Figures 2 and 3.

Figure 2. The green shade is making the green lines harder to read. Suggest changing the shading to grey color. One of the green lines (cyan?) should be changed to another color, like a blue or orange color. Similarly, the two green lines are too similar in Figures 3, 4, 5 and 6, and some of the supplementary figures.

Figure 7 caption, "difference ... between ... (upstream) and ... (downstream)." This can be misleading as if the difference is calculated by upstream minus downstream. Please add a sentence after that, such as "That is, differences are calculated as downstream values minus those in upstream".

Figure 10 caption, 2500 mand, should be 2500 m and.

Figure 10, any description on the white, grey and black lines in the contour plots?

Figures 11, 12 and 13 b, is $T\_min$ the same as $t\_ct$? Please be consistent with the text.

---

## Author Comment (AC1) · 27 Apr 2020

**Reply to Anonymous Referee #2**

*This study presents an evaluation of the Unified Model (with the CASIM module) using wave cloud aircraft observations. Multiple heterogeneous freezing parametrizations are examined and compared with the observations. In addition, a large number of the KiD model simulations are used to examine the impact of the mountain wave period and cloud-top temperature on the cloud evolution. Finally, the authors provide a conceptual model based on the acquired knowledge, which estimates the wave cloud-driven redistribution of water vapor.*
*I find this manuscript quite comprehensive and well written. I like the conceptual model idea, although it is obviously case-dependent (as suggested by the authors as well). However, I think that additional work needs to be done before this manuscript can be published in ACP (Major revisions).*

**Major comments**

- "*UM results: I am not convinced that there is a very good agreement between the UM and the observations (l. 239-244). The statement in the text is subjective, that is, in terms of percentage, the specific humidity errors are very large, on the order of up to several tens of percent (see fig. 2b). Same for the vertical velocity amplitude - errors of 1 m/s can be on the order of ~50% relative to the observations (e.g., leg 2 – fig. 2b). As the UM simulations serve as a benchmark for the KiD simulations, the implications of a weak agreement between the UM and the observations could be significant.*"

**Reply:** It is challenging to take a global model analysis and nest down to 250m resolution to have the waves exactly match 4-5 hours into the simulation. There are limitations imposed by the initial analysis fields, the representation of the orography, drag and dynamics as well as the microphysics. We give numerical values to indicate how close the simulation is to observations To our knowledge this is the first ever study, where such a direct comparison of model and observations has been attempted for orographic wave clouds.

The specific humidity in Fig. 2 is about 10% different for the interpolated values compared to the measured values, which is well within the predicted variability. It is very unlikely for the model to predict the observed humidity at exactly the time and location of the observations. However, the model values taken over the one hour interval around the measurements contain the observed specific humidity values. As the model suggests a quite significant temporal variability of the upstream moisture profile and no continuous information (in time or vertical profiles) is available, it is very hard to speculate how the observed differences affect the condensate content at points inside the cloud. The aircraft measurements do not allow for a quasi-Lagrangian approach, which would be necessary for robust assessment of the modelled condensate content.

Regarding the vertical velocity (Fig. 3), there are differences in the vertical velocity field. Peak-to-peak magnitudes are captured to within 30% and the wavelength appear similar although admittedly harder to quantitatively specify when only a couple of wavelengths are observed. However, the peak velocity is not the most important aspect on its own, rather the absolute height displacement experienced by a parcel (which will be linked to the maximum velocity and wavelength). This is difficult to ascertain for the observations due to only sampling at one level. In addition, aircraft measured vertical velocities can have systematic errors of up to several tenths of a metre per second (see Field et al. 2012: The absolute accuracy of the vertical wind measured from an aircraft is limited by the accuracy at which the aircraft angle of attack and height above ground is known. Typically this would result in a systematic error on the order of tenths of a meter per second). Based on this uncertainty of observations and the uncertainty in numerical model predictions mentioned above deviation between model and observations of around 1 ms$^{-1}$ is judged as very good.

The matching of the humidity curve between the model and observation (Fig. 4a-c) is probably the best test of the combined representation of thermodynamic structure and dynamic evolution of the model. It can be seen in all three passes that the model specific humidity west of 105°E is generally within 10% of the observed value (which itself has an error of about 1-3%), and east of

105°E. where the differences in ice treatment become more important, the modelled specific humidity is within 10-30% of the observed value.

The UM simulations are not strictly used as a benchmark for the KiD simulations. The comparison of the UM to the observations serves only to indicate that the CASIM microphysics seem to be able to roughly capture the general cloud microphysical evolution within the cloud. Bearing the above discussed levels of agreement and the challenges for a more vigorous assessment in mind we believe this conclusion is valid. The KiD simulations in turn are used to expand on the UM simulations in order to sample a larger section of the relevant parameter space. As already pointed out in the conclusion and discussion section a more comprehensive measurement campaign is needed to provide true observational constraints on the UM simulations as well as the conceptual model. This is more prominently highlighted in the discussion section now.

**Changes to manuscript:** We added some extra text including the reasoning above in section 3.1 and 3.2 on why we think the match between UM and observations is good. In the conclusion a paragraph has been added regarding the validity of the conceptual model given the levels of agreement between model and observations as well as on the possibility to obtain more vigorous constraints on the model from observations. (modifications / extra text: lines 250-253, 260-261, 263-273, 664-675)

- "*Homogeneous freezing regime: given the fact that for a large part, the heterogeneous freezing parametrization is examined here, the UM simulations, and hence, also a large fraction of KiD simulations, are "contaminated" by homogeneous freezing in the top of the cloud layer (e.g., fig. 7), which obviously non-linearly impacts the underlying heterogamous cloud layer. The authors did not refer to the homogeneous freezing parametrization in the UM and how well it corresponds with the parametrization in the KiD model. Now, I presume that the UM is too complicated to vary the homogeneous freezing parametrization, but I suspect that the influence of other parametrizations can be examined in the KiD model. An additional approach to address large parts of this comment would be to run the UM initialized without homogeneous freezing influence on the cloud layer (e.g., by offsetting the temperature profile to higher values while retaining the RH profile). I wonder how much would the results of this study change in that case (e.g., deviations of the UM from the observations)?*"

**Reply:** We have now included a reference for the used homogeneous freezing parameterisation in the CASIM description. CASIM is used in the UM and KiD, i.e. both models have exactly the same microphysics representation. Additionally, a new simulation has been performed, in which homogeneous freezing is switched off. As is evident from the results (which are now included), homogeneously formed ice crystals do have no impact on the cloud microphysics in the updraft region of the cloud. However, they significantly contribute to the ice crystal mass and number concentration in the downdraft region (as was already stated in the original manuscript). Hence the main conclusions in the paper regarding the evaluation of heterogeneous freezing parameterisations is valid irrespective of homogeneous freezing. Discussion of this new simulation and the implications have been included in the new manuscript, where appropriate.
Of course, differences in homogeneous freezing will have an impact on the sedimentation fluxes (as will uncertainties in the diameter fallspeed relationship). It is beyond the scope of the present study to investigated the uncertainty of sedimentation fluxes due to these issues. Also sedimentation fluxes cannot be verified with currently available observational data. In so far the KiD model results should only point to interesting parts of the phase-space that should be sampled in future campaigns to provide constraints on sedimentation fluxes. In the conclusions we included a stronger statement alerting readers to these additional sources of uncertainty.
**Changes to manuscript:** modifications / extra text: lines 214-218, 304-307, 320-322

- *Deposition nucleation is not mentioned at all in the text, although the current understanding is that its efficiency significantly increases when we approach the homogeneous freezing regime (e.g., Kanji et al., 2017, https://journals.ametsoc.org/doi/full/10.1175/AMSMONOGRAPHS-D-16-0006.1). I understand that we would typically except immersion freezing to still account for most*

*of the nucleation, but the deposition mode should still be mentioned in the introduction as well as in the model description, even if it is eventually omitted from the simulations.*

**Reply:** Deposition nucleation is currently not represented in CASIM. We added a few sentences on deposition nucleation in the introduction and the CASIM description. However, as the reviewer already states, this nucleation mode is most likely not very relevant in orographic wave clouds.

**Changes to manuscript:** modifications / extra text: lines 64, 79, 210-211

- *Negligible impact of rain processes in the KiD simulations and neglecting rain processes in the conceptual model: the minor impact of rain could be driven by: a. the dominance of the homogeneous ice precipitation from cloud top, b. high cloud droplet number concentration that leads to weak collision-coalescence in the model, and/or c. influence of the collision-coalescence kernel implemented in the model. I am not convinced that rain processes are indeed negligible and whether they should be neglected in the conceptual model. If necessary, I presume that adding a "rain component" to the conceptual model should not be a difficult task, given previous conceptual models for warm clouds (l. 435-436).*

**Reply:** The CASIM microphysics used in both the UM and in the KiD model do include rain formation processes (autoconversion and accretion, following Khairoutdinov and Kogan (2000)). The maximum rain mass mixing ratio in the UM simulations is more than a magnitude smaller than that of either liquid or ice in the UM ($<10^{-5}$ kgkg$^{-1}$ compared to $10^{-4}$ kgkg$^{-1}$) and even smaller than that in the KiD simulations ($<10^{-10}$ kgkg$^{-1}$). In general, bulk microphysics schemes produce rain too early when compared to a detailed size resolved microphysics scheme (Hill et al. 2015). We think the time air parcels spend in the updraft region is too short for significant rain formation. Most air parcels spend less than 15 min in the updraft region (based on the trajectory analysis, not shown). According to considerations of typical timescales for rain production (Seifert and Stevens, 2008; Miltenberger et al. 2015) this is too short for significant rain formation.
We have checked the simulation results again for the impact of rain and cloud droplet sedimentation for the overall downward moisture transport. The contribution from rain sedimentation is below 1% for all cloud top temperatures. However, sedimentation of cloud droplets adds considerably to the total downward moisture transport for cloud top temperatures warmer than -32°C. For colder temperatures homogeneously formed ice dominates the downward moisture transport. As the sedimentation flux for cloud droplets depends only on the advective timescale, i.e. the time period used in the simulations, but if included in the downward moisture transport confuses the fitting of the timescales, we suggest adding this separately.
We have included this discussion in section 4 of the manuscript.

**Changes to manuscript:** modifications / extra text: lines 472-481, 527-540

- *Importance of periods longer than 1000 s (stated in the abstract and l. 554-557): do not believe this conclusion, as the authors did not consider different obstacle heights (eta values I presume). As a result, the impact of the (likely maximum) vertical velocity is nearly ignored in the parameter space evaluation, although it should definitely impact the longevity of the wave cloud, among other factors, via ice processes, especially in the homogeneous freezing regime, where we would expect to lose all condensated droplets relatively fast.*

**Reply:** The in-cloud timescale essentially scales with the upstream flow velocity and the obstacle width if linear gravity wave theory is used (e.g. Miltenberger et al. 2015). A timescale 1800s corresponds to a mountain width of 18 (54) km assuming a horizontal wind speed of 10 (30) ms$^{-1}$. Note we are considering here mid-level, isolated wave clouds and not thick orographic clouds producing significant surface precipitation (in contrast to previous conceptual models for orographic precipitation such as e.g. Smith and Barstad 2004 or Miltenberger et al. 2015). For the former typically the width of single mountains is essential, while for the latter typically the width of the mountain range, instead of the width of isolated mountains is more representative. For lee-wave clouds typical wavelength reported in literature are shorter than 20 km (Grubisic et al. 2008), i.e. are well covered by the time period range investigated here. For cap clouds there are to our knowledge no estimates available, but the likely spatial extend estimated from photographs is also on the order of 10-50 km.

Of course, for clouds containing ice crystals (or other hydrometeors with long evaporation timescales) the mountain height influences the cloud extend by effecting how much water is condensed and how long the ice crystals can survive in sub-saturated regions. This effect is independent of the wavelength (or time period of the wave clouds). Although it would be interesting to investigate this, systematically investigating this effect would require a substantial amount of additional simulations (at least doubling the 45000 simulations already analysed). This is beyond the scope of the present paper.

We have included a paragraph in the description of the KiD set-up as well as in the discussion section pertaining to this issue.

**Changes to manuscript:** modifications / extra text: lines 172-180, 644-650

**Minor comments:**

- *l. 49-52 – That is a rather complex sentence. I suggest rephrasing or breaking into two separate sentences.*

  **Reply:** The sentence has been split into three sentences.

- *l. 109 – ICE-L acronym definition is missing.*

  **Reply:** added.

- *l. 109 - please add parentheses to the citations.*

  **Reply:** done

- *l. 109 – inconsistent date (November 17th) with the following sections and figures.*

  **Reply:** corrected. Thanks for spotting this.

- *l. 127-130 – These two sentences seem redundant - repeating information already provided in the Introduction.*

  **Reply:** removed.

- *l. 146 - Suggest consistency in the number of fractional coordinate digits.*

  **Reply:** done

- *l. 147 – focusses –> focuses*

  **Reply:** corrected

- *l. 167 - Please define the source for this eta value - I will presume that it is equivalent to the obstacle (mountain) height*

  **Reply:** This value is based on the mean maximum η value of trajectories in the UM model, which pass through the wave cloud. A sentence stating this has been added. η is roughly equivalent to the obstacle height close to the obstacle, but varies with height above ground according to the vertical structure of the gravity wave.

- *l. 168 – 32.1 K - Suggest consistency about the temperature units (C instead of K).*

  **Reply:** Sorry for the confusion. We have carefully checked (and corrected) the temperature units in the manuscript.

- *l. 173 – Eq. 2 I do not find consistency in the definition of the different z ranges.*

  **Reply:** corrected.

- *l. 175 – (nothing to revise here) – Figure 9 is very nice.*

  **Reply:** Thank you very much :)

- *l. 201-202 – What is the parametrization for ice hydrometeor fall velocity used in UM and KiD? This may have a substantial impact on the results presented below.*

  **Reply:** Both models use a (the same) relation between ice diameter and fallspeed. The equations and parameters are now provided in the CASIM description. We added also a comment on the sensitivity of sedimentation fluxes to the parameterisation of fall velocities.

- *l. 242 – I presume "basis" should be "bias"*

**Reply:** corrected.

- *l. 252-253 - If the correction is made for 0.02 g/kg, what is the importance of 0.0001 g/kg in fig. 1?*

**Reply:** Thanks for spotting this inconsistency. The new Fig.1 shows now only qi / ql larger than 0.02 g/kg.

- *l. 282 - I can only see some sort of an ni agreement in fig. 5f. I can't interpret the max ni intersection in panel e as model-observations agreement. Using that parameter as a measure of model performance could be misleading.*

**Reply:** We added a sentence to better describe the agreement (or lack thereof) in ice crystal number concentrations

- *l. 321 - "data .. is . . ." - "Data" is the plural form of "datum" - please correct the text and figure captions accordingly (is –> are, etc.)*

**Reply:** corrected.

- *l. 330 - best agreement with DM10 and TB13 - how do you define the best agreement? In some aspects (e.g., absolute ni values), the log scale in fig. 6 is misleading because I would suggest that the best agreement is with DM10 and DM15.*

**Reply:** You are right. We have expanded the text on this issue and also include the DM15 parameterisation.

- *l. 365 - 0.1 g/kg - these are mixing ratio units, not flux units. I suggest providing a more consistent terminology throughout the text (also related to the conceptual model).*

**Reply:** Yes, you are right. These are the Lagrangian integral of the sedimentation fluxes. We have changed the terminology everywhere in the manuscript (using instead: total downward moisture transport).

- *l. 403 - a minus sign is missing for the temperature*

**Reply:** corrected

- *l. 406 - I suggest adding a reminder to the reader about which two simulations are discussed here.*

**Reply:** We are here not referring to two simulation in particular. For each wave period, cloud thickness, and cloud top temperature combination there are 20 simulations with different settings in the cloud microphysics. We compute the difference between any combination of these 20 simulations and show the maximum difference from these in Fig. 10b. We have rephrased the sentence to make this clearer.

- *l. 445-446 - I presume that the definition of Gpot is the integration of qv minus qs(Tmin), yes? Please clarify, or alternatively, describe eq. 4 earlier in the text.*

**Reply:** Gpot is qv -qs (Tmin), no integration needed. We have added a sentence at the beginning of the paragraph to make this clear.

- *l. 473 " –saturation)" - redundant parenthesis*

**Reply:** corrected.

- *l. 479 – inline equation - not all terms in this equation are defined.*

**Reply:** added.

- *l. 492 – pre-scribed –> prescribed*

**Reply:** corrected.

- *l. 500-501 & Fig. 13 - I suggest adding a panel that shows the difference between the conceptual model and the KiD model. The discrepancies I see in Figure 5 in the SI (especially in panels c and d, which are shown in log scale) suggest that the differences can be quite significant when the time period and Tct parameter space are examined. Also, shouldn't there be consistency re-*

*garding the discussion/use of total water (qt) and water vapor (qv) throughout the text in figures (7-13)?*

**Reply:** The relative difference between the conceptual model and the KiD model are shown in Fig. 13b. Errors are up to 30% of the downward flux. Also, we checked the use of total water and water vapour for consistency.

- *l. 504 - I presume that 'b' is missing when referring to fig. 13.*

  **Reply:** yes, corrected.

- *l. 538 - Except for the conceptual model, I did not encounter any discussion about the results based on varying cloud thickness. I suggest adding some information to the text and figures, or removing it from this discussion about the conclusions.*

  **Reply:** The KiD results are only shown for cloud thickness of 2 km, which roughly corresponds to the ICE-L cloud. This is mentioned in the figure captions. We added a reference to the cloud thickness, when discussing the Fig. 10.

- *Fig. 1 - Suggest changing the cyan, blue, and gray curve colors - they mixed with the colormap. Also, the units in the colorbar are confusing here. Also, What altitude do the black contours represent? This should be specified.*

  **Reply:** todo

- *Fig. 1 and discussion in the text - Is 10ˆ-7 kg/kg above the aircraft instrumentation uncertainty level for IWC and LWC? This should be discussed and justified in the text, i.e., what is the "true" extent of this cloud field given measurable justified quantities?*

  **Reply:** Thanks for spotting this inconsistency. The new Fig.1 shows now only qi / ql larger than 0.02 g/kg.

- *Fig. 2 - I suggest redefining the altitudes for each flight leg*

  **Reply:** Not sure what you mean here? Based on the comments from reviewer 2, we have added labels (A, B, C) to the different flight legs, which are used throughout the text and the altitude of which is specified in the text and the Fig. 2 caption.

- *Fig. 4 - I'm having a tough time reading the axes labels*

  **Reply:** enlarged.

- *Fig. 3-5 - please provide a title for each panel stating the flight leg and/or altitude. At the moment, it is hard to follow the text.*

  **Reply:** done.

- *Fig. 5-6 – suggest adding a legend instead of directing the reader to fig. 4 every time.*

  **Reply:** done

- *Fig. 7 - blue for positive values in panels a,b is counter-intuitive. I suggest flipping this colormap.*

  **Reply:** We find the colourmap intuitive as is, as positive values mean moistening (blue) and negative values drying of the air parcels (red). We therefore decide not to implement this suggestion.

- *Fig. 9 - Please correct qv–>qt in the legend*

  **Reply:** done

- *Fig. 10 - The figure caption is not complete, e.g., last sentence, circle markers, the definition of the two simulations, etc. The varying contour colors in panels b,d are quite confusing.*

  **Reply:** We reformulated and added additional information to the figure caption. We have deliberately chosen different colour scales to alert the reader that one column of plots (a,c) are showing absolute (mean) values, whereas the right column plots (b,d) are shown the spread of the mean values. As this was intentional and we still believe using different colormaps is meaningful, we refrain from implementing the change in colourmap.

---

## Author Comment (AC2) · 27 Apr 2020

**Reply to review by Minghui Diao**

*This manuscript uses the ICE-L field campaign measurement to compare with simulations from the Unified Model. In addition, idealized simulations from a 2-D model, the KiD model, were used to conduct additional simulations and examine downward moisture flux. The UM model included a recently developed module – CASIM, which enables the analyses of dust particles and their impacts on liquid and ice hydrometeors. The overall organization of the writing is straightforward. The sensitivity tests on various heterogeneous nucleation parameterisations provide valuable information. The reviewer has a few major comments, followed by some minor comments. A major revision is recommended before being considered for publication at ACP.*

1. *For the ICE-L field campaign, the 2DC data are restricted to > 125 micron. However, it is not clear if the model outputs of ice water content (IWC) have considered the size cutoff in ice crystal size distribution. From Figure 5 axis label and caption, it seems that ice crystal number concentration (Ni) has been restricted to > 125 micron. But in the legend of Figure 4, qt did not mention any size cutoff. Please clarify this in the main text besides the figure legend.*
   *Also, can the authors comment on possible impacts of the comparison results if small particles were included in the comparison? For example, would the model show better or worse results compared with observations?*

   **Reply:** The measured ice water contend comes from the integrated 2D size distributions for sizes >125 µm. After some discussion with Andrew Heymsfield and consideration of additional available data from the 2DS instrument, we have now included data for crystals down to a size of 50 µm. This is possible for the ICE-L data as the impact of shattering in the lenticular wave clouds is probably minimal due to the overall small ice crystal sizes. The validity of 2DC data for small ice crystals in the ICE-L data-set is supported by the agreement with 2D-S data, that has a lower detection limit.

   The modelled values have been derived by integrating over the ice particle distribution, which, however, is not explicitly represented in the model. The modelled size distribution is derived from the ice mass mixing ratio, number concentration and assumed shape parameter. For the number concentrations this had already be done in the original manuscript. We have now taken the same approach for the ice mass mixing ratio (which is also used to correct the total condensate). Differences are generally very small.

   If there were a significant amount of ice crystals smaller than 125 (50) µm present in observations (and not in the model), the comparison to modelled number concentration would improve in the region, where heterogeneous freezing dominates (west of ~ -105.1°W). In the part of the cloud influenced by homogeneously formed ice crystals, the model would compare less well with the observations in terms of ice water content and number concentrations for flight legs B and A (improved fro flight leg C). Underestimating the presence of small ice particles affects the comparison of number concentration much more than that of ice water content.

   **Changes to manuscript:** We updated Fig. 4 d-f and Fig. 5 a-c using the corrected model ice water content values (including additionally 2D-S data and using a threshold of 50 µm). We now also discuss the implications for the comparison from the measurement restrictions to larger ice particles in section 3.2.
   modifications / extra text: lines 345-356

2. *Another main comment is related to measurements of 2DC and CDP. In Line 133 – 135, the authors commented that 2DC probe is used for IWC and Ni, and CDP is used for cloud number concentration (I assume that you mean liquid droplet number concentration?). However, in previous studies, we found that 2DC may measure some large drizzles, while CDP may measure some small ice. A detailed discussion about separating liquid and ice from 2DC and CDP measurements was given in D'Alessandro et al. (2019, J. Climate), https://doi.org/10.1175/JCLI-D-18-0232.1, "Cloud Phase and Relative Humidity Distributions over the Southern Ocean in Austral Summer Based on In Situ Observations and CAM5 Simulations". Can the authors comment on the potential impacts on the model evaluation, if some ice was misidentified as*

*liquid in CDP measurements, and some liquid was misidentified ice in 2DC measurements? Some sensitivity tests on possible variations of IWC, LWC, Nice and Nliq derived from field observations would be helpful.*

**Reply:** Drizzle drops form through droplet coalescence. Drizzle droplets are highly unlikely to form in lee wave clouds at this temperature over the timescales for a parcel to transit through the cloud. Therefore the 2D-C is unlikely to detect liquid for this case. Detection of ice by the CDP is more likely, however the number concentrations will be ~1000 times less than the droplet concentrations and so will not contaminate the droplet number concentration.

**Changes to manuscript:** We added some text to alert the readers to the possible misclassification of ice as liquid particles and vice versa.
modifications / extra text: lines 345-356

3. *Homogeneous freezing has been briefly mentioned in a few places, but there are not many discussions on the quantitative impacts from it compared with heterogeneous nucleation. For example, even though homogeneous freezing is more dominant at colder temperatures, such as at below -37 C, ice crystals formed by homogeneous freezing can sediment into lower altitudes, and therefore being misidentified as ice formed via heterogeneous nucleation. Can the authors comment more on this sedimentation effect?*
   *The reviewer suggests that the authors quantify IWC and Ni into two categories – those originated from homogeneous freezing versus those from heterogeneous freezing. Would this be possible for the model used here? For example, additional lines can be added to Figure 4 (d-f) analysis of qt and Figure 5 (d-f) analysis of Ni, to quantify these two components.*

   **Reply:** Ice crystals formed by heterogeneous nucleation indeed can influence ice cloud properties at lower levels by sedimentation. This is why we limited the comparison of Ni to the heterogeneous freezing parameterisations in Fig. 6 to the cloud part at the upstream edge (west of -105.15°E). Unfortunately it is not possible to trace ice formed by different processes in the UM-CASIM model. However, we have conducted an additional test simulation, where we have switched of homogeneous freezing (using DM10 for heterogeneous freezing). The results from this simulation are now included in Fig. 4 and 5. In the region west of -105.15°E the ice number concentration, cloud droplet number concentration, and cloud / ice mass mixing ratio are identical to the simulation with homogeneous freezing. This justifies our approach taken in Fig. 6. Some discussion has been added regarding the impact of homogeneous freezing on the cloud region further downstream.
   **Changes to manuscript:** modifications / extra text: lines 214-218, 304-307, 320-322

4. *In the conclusion and the result section, when comments were made on whether the model performance is good or not, it seems a little arbitrary. One suggestion is to add some comparisons with previous studies, or with the older versions of the same model. If improvements are seen compared with previous work, then it is more convincing that this model performs better.*

   **Reply:** see reply to major issue 1 from RC2.

   **Changes to manuscript:** see reply to major issue 1 from RC2.

**Minor comments**

- *Line 199, "the he modification". Typo.*

  **Reply:** Thank your for pointing this out. Correction done.

- *Line 161, recommend adding a full description of notations for temperatures and altitudes used in this study. For example, there is t_ct for cloud top temperature, but later in Figure 10, the axis label uses T_min for cloud top temperature. Please be consistent. Cloud thickness is defined as z_c, but the definitions of z_ct and z_cb is not explicitly mentioned (I assume they are cloud top and cloud base height, respectively).*

  **Reply:** Thank your for pointing this out. Correction done.

- *In equation (2), there are notations of z_ct,t and z_cb,t. How are they different from z_ct and z_cb?*

  **Reply:** Thank your for pointing this out. Correction done.

- *Line 168, "32.1 K" should be in Celsius.*

  **Reply:** Thank your for spotting the errors in the units. Corrections done.

- *Line 243, "temperature basis", biases?*

  **Reply:** corrected

- *Line 275, the authors mentioned that "while significant cloud glaciation also only occurs in the downdraft region, ice crystal number concentration increases further downstream". Is there any explanation why significant cloud glaciation only occurs in the downdraft region? It seems counter intuitive that downdraft leads to glaciation and new ice crystal formation.*

  **Reply:** This is the impact of sedimentation from the homogeneous freezing region as evident from the difference between the simulations with and without homogeneous freezing parameterisation. This was already state in the sentence you comment on here. We have rephrased the sentence to make it clearer and also refer to the new simulation without homogeneous freezing.

- *Line 277, "in the model the air parcels likely experience larger vertical displacement", is there any evidence of the parcel displacement? Is it possible that other factors could lead to higher ni, such as homogeneous freezing is being activated too early, allowing too little clear-sky ice supersaturation?*

  **Reply:** There is no direct measure of vertical displacement in the observations, as this is a Lagrangian measure. Deviations between the parameterised and actual homogeneous freezing can lead also to the observed discrepancies. We included this in the text.

- *Line 278, "ice crystal population at observed along flight legs", delete "at observed"? In the same line, "a earlier", an earlier.*

  **Reply:** corrected

- *Line 279, "... ice crystal number masking the depositional growth", should it be "ice crystal number <and> masking the depositional growth"?*

  **Reply:** reformulated to make sentence clearer.

- *Line 292 - 293, "the longevity of ice crystal... related to smaller average ice crystal mass. . .", what is the meaning of ice crystal mass? Do you mean the mass of individual ice crystals, or the total ice water content?*

  **Reply:** Ice crystal mass refers to the mass of the average ice crystal (as calculated based on the assumed distribution and the prognostic variables of ice number concentration and mass mixing ratio). We clarified this in the text.

- *Line 298 – 299, "the overestimation in initial ice crystal number is either related to the heterogeneous freezing parameterisations used or a too large diameter of the newly formed ice crystals." Heterogeneous freezing generally forms fewer ice crystals than homogeneous freezing. Is it possible that the high ni here is contributed by homogeneous freezing? In addition, the comment on the model having too large ice crystals and therefore overestimating Ni doesn't seem right. If the diameters of the newly formed ice crystals are too large, they would sediment faster and reduce the ice crystal number concentration. In addition, if the total water content is conserved, forming too large ice crystals would lead to fewer ice crystals, not more ice crystals.*

  **Reply:** To clarify the impact of homogeneous freezing, we have conducted an additional simulation, in which homogeneous freezing is switched off. It is evident from comparing this to the existing simulations that there is no impact of homogeneously formed ice crystals on the ice crystal number concentration in the part of flight track we are discussing here.
  As to the argument with ice crystal size: The observations only measure ice crystals larger than 125 μm. Newly formed ice crystals can be smaller and hence the first detection of ice crystals

does not only depend on heterogeneous nucleation but also vapour deposition. If in the model, newly formed ice crystals are larger than in reality and we apply the same threshold of 125 µm, crystals will be detected earlier, i.e. closer in time to the nucleation event. Hence, the number of ice crystals will appear larger than in the observations, as a larger fraction of the heterogeneously formed crystals exceed the 125 µm threshold and are detected. We have added a sentence to clarify our argument.

- *Line 305, "observations if", observations of?*
  **Reply:** corrected

- *Line 311, "This data". Data should be in plural form. This typo occurs in several places, including figure captions and the "data availability" section. Please use a global search to correct them all. Same for Line 321, observation data . . . is, should be are.*
  **Reply:** corrected

- *Line 328, "but introduces", and introduces?*
  **Reply:** corrected

- *Line 330, here both DeMott et al. (2010) and Tobo et al. (2013) are mentioned as the ones giving the best agreement. But in the conclusion section, only DeMott (2010) is mentioned. Maybe the conclusion can provide more comments on the best agreement based on specifically what variables.*
  **Reply:** We have expanded the text on this issue and also include the DM15 parameterisation.

- *Line 398, -45 deg c, "c" should be C.*
  **Reply:** corrected

- *Line 403, 37 deg C should have a minus sign.*
  **Reply:** corrected

- *Line 451, the equation (t + A\*gamma) should be (t0 + A\*gamma)? If not, what is "t" here?*
  **Reply:** corrected

- *Line 459, please add a comma between "clouds" and "reflecting". Some other sentences are too long as well without a comma to separate different parts of the sentences.*
  **Reply:** done. We checked the manuscript again and added commata where we deem them appropriate.

- *Line 464, several log_10 didn't have the 0 in subscript.*
  **Reply:** corrected

- *Line 476, w = 0 ms, should be m s-1.*
  **Reply:** corrected

- *Line 479, please clarify the meaning of each term in the equation.*
  **Reply:** added

- *Equation 6. K should be deg C*
  **Reply:** corrected

- *Equation 7. K should be deg C. Also, there is a km unit. Should be C?*
  **Reply:** corrected

- *Line 511 – 512, this would be a good place to add comments on previous model evaluation studies and compare with the results shown here.*
  **Reply:** To our knowledge this is the first study, where a model simulations of mixed-phase orographic wave clouds are directly compared to observations. So this is unfortunately not possible.

- *Line 512, 1 ms, should be 1 m s-1.*

**Reply:** corrected

- *Line 539, -30 K and -40 K, should be deg C.*

  **Reply:** corrected

- *Figures 1, 2, 3. Suggest adding labels to three segments as A, B, C, and use texts and arrows to highlight them in Figure 1b. It would make it a lot easier to match them with the figure legends and lines in Figures 2 and 3.*

  **Reply:** done as suggested.

- *Figure 2. The green shade is making the green lines harder to read. Suggest changing the shading to grey color. One of the green lines (cyan?) should be changed to another color, like a blue or orange color. Similarly, the two green lines are too similar in Figures 3, 4, 5 and 6, and some of the supplementary figures.*

  **Reply:** done.

- *Figure 7 caption, "difference . . . between . . . (upstream) and . . . (downstream)." This can be misleading as if the difference is calculated by upstream minus downstream. Please add a sentence after that, such as "That is, differences are calculated as down- stream values minus those in upstream".*

  **Reply:** A sentence has been added for clarification.

- *Figure 10 caption, 2500 mand, should be 2500 m and.*

  **Reply:** corrected

- *Figure 10, any description on the white, grey and black lines in the contour plots?*

  **Reply:** We noticed these are more confusing then helping so the contourlines have been removed in all plots.

- *Figures 11, 12 and 13 b, is $T_{min}$ the same as $t_{ct}$? Please be consistent with the text.*

  **Reply:** Figure labels have been changed to be consistent with the text.

---

## Referee Report (RR1)

Minghui Diao

The authors have adequately addressed the comments listed in my original review. The reviewer recommends the manuscript being accepted for publication in ACP.